# Topological regression as an interpretable and efficient tool for quantitative structure-activity relationship modeling

Ruibo Zhang[1,3], Daniel Nolte[1,3], Cesar Sanchez-Villalobos [1], Souparno Ghosh[2] ✉ & Ranadip Pal [1] ✉

Quantitative structure-activity relationship (QSAR) modeling is a powerful tool for drug discovery, yet the lack of interpretability of commonly used QSAR models hinders their application in molecular design. We propose a similarity-based regression framework, topological regression (TR), that offers a statistically grounded, computationally fast, and interpretable technique to predict drug responses. We compare the predictive performance of TR on 530 ChEMBL human target activity datasets against the predictive performance of deep-learning-based QSAR models. Our results suggest that our sparse TR model can achieve equal, if not better, performance than the deep learning-based QSAR models and provide better intuitive interpretation by extracting an approximate isometry between the chemical space of the drugs and their activity space.

Quantitative structure–activity relationship (QSAR) models have become an essential tool in pharmaceutical discovery, especially in the virtual screening for hits and lead optimization stages[1]. Experimental characterization of candidate molecules is expensive and time-consuming. As a relatively easy-to-implement alternative, QSAR models could be a valuable tool for assisting chemists by providing design ideas to prioritize their experiments. QSARs are usually supervised machine learning models that describe the connections between chemical structures and their biological activities, such as their potency, physicochemical properties, pharmacokinetic properties, or environmental effects[2]. QSAR models enable in silico structural design by providing property predictions from machine-readable representations of the chemical structure, thereby helping generate and prioritize design ideas. This technique has been widely applied in virtual screening and lead optimization with a fair amount of success[1,3].

In QSAR methods, chemical substances must first be transformed into machine-comprehensible mathematical representations. Three commonly used representations are (a) vectors such as classical molecular descriptors or molecular fingerprints (FPs), (b) graphs, and (c) strings such as Simplified Molecular Input Line Entry System (SMILES). Classical molecular descriptors[4] encode a specific computed

or measured attribute of the molecule into a single number, for instance, the count of bonds, atoms, functional groups, or physico-chemical characteristics, and are often used in combination to form feature vectors. PaDEL[5], Mordred[6], and RDKit are examples of popular descriptor-calculation software packages for numerically representing the chemical structure and molecular characteristics. Extended-connectivity fingerprints (ECFPs)[7] are an example of a topological fingerprint computed using a variant of the Morgan Algorithm that encodes chemical substructures by atom neighborhoods using a high-dimensional sparse bit-string representation. The graph representation, on the other hand, characterizes 2D chemical structures as graphs, with atoms as vertices and bonds as edges. SMILES specify a notation for representing the chemical graphs of molecules as strings of characters.

Once the chemical structures are represented using a suitable protocol, a predictive method is chosen to connect the structural information with the functional properties. For instance, if the chemical structures are represented as strings or graphs, deep-learning methods are often used for prediction due to their ability to perform embedded feature extraction. Chemprop[8], in particular, has turned out to be a popular method that uses directed message-passing neural

[1]Department of Electrical and Computer Engineering, Texas Tech University, Lubbock, TX 79409, USA. [2]Department of Statistics, University of Nebraska - Lincoln, Lincoln, NB 68588, USA. [3]These authors contributed equally: Ruibo Zhang, Daniel Nolte. ✉e-mail: sghosh5@unl.edu; ranadip.pal@ttu.edu

networks to learn molecular representations directly from the graphs to predict the properties of molecules. This method has been shown to excel at antibiotic discovery[9,10] and lipophilicity prediction[11] indicating its potential as a QSAR model. With the rise in popularity of large language models and the attention mechanism, the use of SMILES strings has been increasingly investigated for their potential embedded feature extraction, predictive performance, and interpretability. For example,[12] pre-trained a transformer-based network through masked SMILES recovery, and offered the pre-trained model for transfer learning onto specific tasks. Similarly, Transformer-Convolutional Neural Network (CNN)[13] applied the transformer architecture to canonicalize SMILES string inputs and enables transfer learning of the model onto specific activity prediction tasks.

QSAR models are often developed for their predictive performance. However, the effectiveness of QSAR models, as a computational tool assisting molecular discovery and design, could be greatly improved by enhancing their domain-specific interpretability. Model interpretability, usually defined as the ability to explain predictions in a human-understandable way[14], typically consists of computing feature importance scores[15–18], influence functions to identify training instances most responsible for the prediction[19], developing locally interpretable models to approximate global black-box algorithms[20–22], and generating counterfactuals[23,24]. For example, standard shallow learners, like Random Forests (RF) and Support Vector Machines (SVM) are often used in QSAR modeling to offer feature importance scores[25]. However, molecular interpretability is largely based on the interpretability of the underlying molecular representation. For instance, ALogP can be used as an important classical descriptor that plays a key role in determining the solubility of a molecule. However, a target value of ALogP cannot be mapped back to a precise chemical structure. When using interpretable fingerprints, the foregoing feature importance scores could potentially map prediction contributions onto the molecule to visualize which substructures positively or negatively impacted the prediction[25–27]. Although feature importance measures increase the explanatory power of machine learning models, caution must be taken when these scores are invoked on molecules outside the applicability domain of the model, as prediction importance does not always translate to biological relevance[28]. Locally interpretable models can be fitted to explicate predictions of black-box models. For instance, SHapley Additive exPlanations (SHAP) offers a model-agnostic method for calculating prediction-wise feature importance[21,22]. Since this technique usually informs which features contributed to the specific test instance's model prediction the most, it may not always lead to actionable design ideas. Thus[24], proposed Molecular Model Agnostic Counterfactual Explanations (MMACE) to generate counterfactual explanations that would help answer the question: what changes will result in an alternate outcome, regardless of the underlying model used. These methods are based on the model's knowledge and, therefore, may be influenced by chance correlation, rough response surfaces, and overfitted models, leading to disappointing results[29]. Recent advances in the attention mechanism of deep learners offer some explanatory power[30]. For instance[31], uses Layer-wise Relevance Propagation to provide structural interpretation of nodes and edges (atoms and bonds), Transformer-CNN incorporates Layer-wise Relevance Propagation to calculate individual atom contributions influencing the predictions, and[32] uses salient maps to highlight the substructures closely related to the model output. These maps are analogous to the foregoing feature importance concept and have similar drawbacks in terms of deriving actionable insights for the design of new molecules.

Similarity-based methods[33] (k-nearest neighbor (KNN), kernel regression[34,35], and pairwise kernel method[36]), provide natural intuitive interpretation at the instance-level by directly providing the training instances that influenced the model's prediction the most. For example, read-across is a popular alternative property prediction

technique that finds the most similar chemicals to the query chemical. Numerous publicly available tools use some variants of read-across techniques to aid chemists with design ideas[37]. These tools allow chemists to assess the potential of the selected analogous neighbors to infer properties of the query chemical. Additionally, similarity-based methods allow informative visualizations through network graphs derived from the similarities. Network-like Similarity Graphs (NSG)[38] were developed to guide lead optimization in drug discovery and have often been used to display the complex activity landscapes and the relationships between chemicals within a target set in 2D. Expanding this to drug-target interactions, methods like Similarity Ensemble Approach (SEA)[39] and Chemical Similarity Network Analysis Pulldown (CSNAP)[40] enable visualization of drug-target interaction networks and the prediction of off-target drug interactions, which have led to deeper investigations into drug polypharmacology and the discovery of off-target drug interactions[41,42]. As we show later, these chemical similarity networks allow the clustering of similar molecules, which enables practitioners to mine regions of desired activity for innovative design ideas and potential leads. In addition to providing prediction-wise training instance importance, these graph structures are directly compatible with Laplacian Scores[43] for global feature importance, which have been used in QSAR modeling for feature selection[44,45]. Since SHAP and MMACE are model agnostic, they can also be paired with similarity-based QSAR models to allow prediction-wise feature importance and the generation of unseen counterfactuals. Thus, similarity-based methods can provide multiple layers of interpretability on top of the commonly applied chemical similarity interpretation and visualization methods listed above.

However, a problem in similarity-based QSAR is that most QSAR methods assume that similar structures lead to similar activities, which is often violated in chemical structure modeling due to the prevalence of *activity cliffs* (ACs)[46], which are pairs of compounds with similar molecular structures, but with a large difference in potency against their target[47]. The existence of ACs often cause QSAR models to fail, especially in the lead optimization stage[48], and limit the prediction performance across the drug landscape, leading to the use of network-based methods to interpret and analyze their behavior[38,49]. One way to use similarity-based methods in the presence of ACs is to learn the similarity metric from the data itself, instead of choosing a similarity metric a priori. Large margin nearest neighbor[50] is a very popular algorithm for supervised metric learning when the response variable is categorical. For continuous response variables, Metric Learning Kernel Regression (MLKR)[51] is perhaps the most popular algorithm to estimate the similarity metric. Metric learning techniques offer good explanatory power because once the metric is learned, the chemical space of molecules is approximately isometric to the activity space, resulting in smoother structure–activity landscapes as shown in[52]. Consequently, under the learned metric, high-activity molecules are clustered relatively tightly in the chemical space and therefore, that space could be mined for new molecules. Figure 1 depicts this phenomenon using various projection methods Generative Topographic Mapping (GTM)[53], Multidimensional Scaling (MDS), t-distributed Stochastic Neighbor Embedding (t-SNE), and Uniform Manifold Approximation and Projection (UMAP), to show the interpolated activity landscapes of the protein target Coagulation factor XIII, or CHEMBL4530, in 2 dimensions and compare them with an MLKR-based representation. Observe that, except MLKR, none of the other methods were able to separate two chemically-similar-but-functionally-different molecules, CHEMBL208650 and CHEMBL2086502, which have a Tanimoto similarity between ECFP4 fingerprints of 0.70 but target difference of 2.61. This is to be expected because, in their original form, GTM, MDS, t-SNE, UMAP are all unsupervised techniques and do not incorporate the activity information in their projections. MLKR, on the other hand, is a supervised metric learning method,

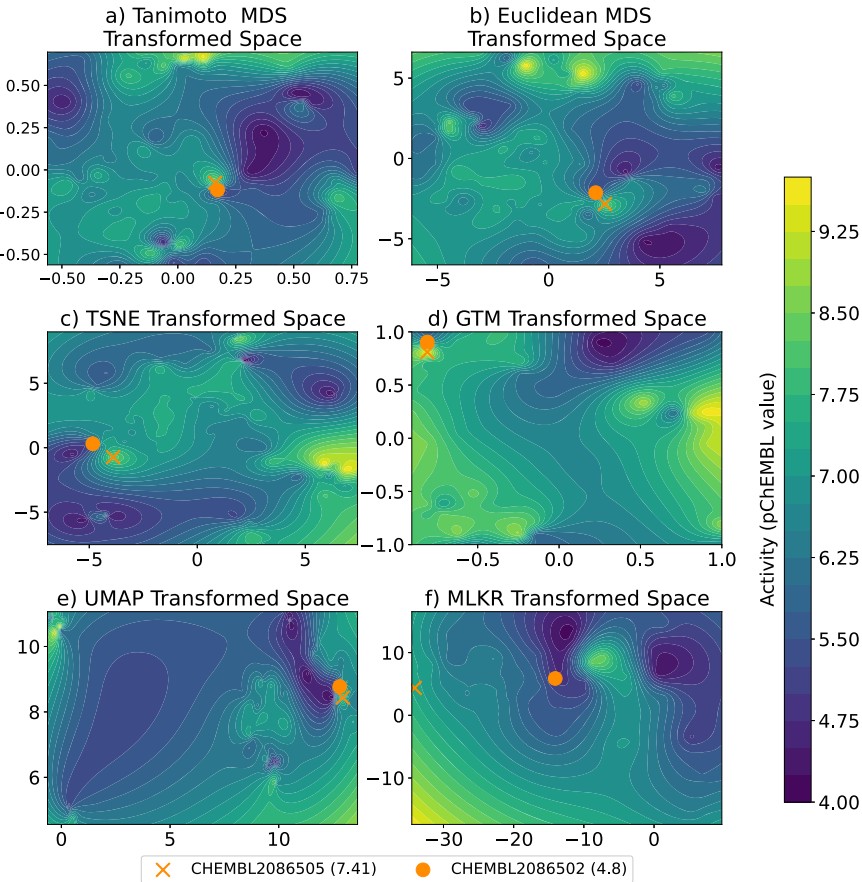

**Fig. 1 | The transformed chemical space and interpolated activity surface of target CHEMBL4530 using various projection methods.** Transformed and interpolated chemical space using (**a**) multidimensional scaling (MDS) with Tanimoto distance, (**b**) MDS with Euclidian distance, (**c**) t-Distributed Stochastic Neighbor Embedding (TSNE) using Euclidian distance, (**d**) Generative Topographic Mapping (GTM), (**e**) Uniform Manifold Approximation and Projection (UMAP) using Euclidian distance and (**f**) Metric Learning for Kernel Regression (MLKR). Notice how activity cliffs are present regardless of the projection method. Additionally, notice that MLKR creates the smoothest activity space and best separation of the two similar (Tanimoto Similarity between ECFP4 fingerprints = 0.70) molecules, CHEMBL2086505 (pChEMBL = 7.41) and CHEMBL2086502 (pChEMBL = 4.8).

which allows it to incorporate the target activity information resulting in smoother activity landscapes.

In this paper, we develop an MLKR-inspired regression-based technique, topological regression (TR), that models the distance in the response space using the distances in the chemical space. TR essentially builds a parametric model to determine how pairwise distances in the chemical space impact the weights of nearest neighbors in the response space. Observe, unlike metric learning techniques, TR does not attempt to learn a metric in the response space, nor does it attempt to provide a lower dimensional projection like MDS or GTM. Rather, TR simply estimates the weights of nearest neighbors. In comparison to traditional modeling methods, like RFs and SVMs, which are dependent on a predefined fingerprint, TR can accommodate non-metric systems and does not crucially require coordinates for each instance. As we will show in the subsequent sections, TR can work on the similarities between training molecules, such as those computed from molecular kernels[54,55], thereby circumventing the problem of featurization of molecules. Since, our primary use-case scenario is QSAR in the lead identification/optimization process, where the contiguity of high-activity molecules plays a significant role, we perform a large-scale comparison on 530 ChEMBL bio targets. We use RF, ChemProp, and Transformer-CNN as baseline models and show that TR matches the performance of Transformer-CNN at a significantly less computational cost. We also observe, empirically, that TR produces numerically superior predictive performance as compared to the other competing methods. Additionally, both MLKR and TR produce reasonably

contiguous areas of high activity, thereby identifying a relatively compact high-activity chemical space.

## Results

### Model performance comparison on ChEMBL datasets

We apply our TR method with Gaussian kernel neighbor weighting on 530 ChEMBL datasets under both random split and scaffold split. As explained earlier, we use the ECFP4 TC distance as input to TR to predict the activity values. We use 80% of all the instances in each dataset for training and the remainder for testing. For the construction in the section "multivariate construction of topological regression", when $I' \cap I = \phi$, we use 20% of the training instances as anchor points and the remaining 80% of the training set for neighborhood training. We denote this method as TR* in the results. For the approach described in the section "univariate construction of topological regression" without disjointedness requirement, we use 50% of training instances, with a maximum of 2000 instances to improve computation time, as anchor points, and those results are denoted as TR. Finally, to reduce the sensitivity of results to anchor point selection, and to improve generalization error, different random sets of anchor points were sampled to create an ensemble of TR models(see the section "ensemble topological regression"). We denote this method as Ensemble TR and used $t = 15$, $\mu_k = 0.6$, and $\sigma_k^2 = 0.2$ for the subsequent results.

The average Spearman correlation and NRMSE for each method (RF, MLKR with KNN, ChemProp, TCNN, TCNN with augmentation,

**Table 1 | Comparative measurements obtained on each of the competing methods**

| | CV Split | | Scaffold | |
|---|---|---|---|---|
| | Spearman | NRMSE | Spearman | NRMSE |
| RF | 0.7629 | 0.6242 | 0.6493 | 0.7395 |
| MLKR | 0.7486 | 0.6421 | 0.6367 | 0.7593 |
| ChemProp | 0.7160 | 0.6776 | 0.5986 | 0.8002 |
| TCNN | 0.7437 | 0.6595 | 0.6321 | 0.7692 |
| TCNN Aug | **0.7858** | **0.5961** | **0.6742** | **0.7176** |
| TR* | 0.6935 | 0.7023 | 0.5793 | 0.8174 |
| TR | 0.7625 | 0.6255 | 0.6531 | 0.7358 |
| Ensemble TR | **0.7847** | **0.5989** | **0.6791** | **0.7101** |

Normalized root mean square error (NRMSE) and Spearman's correlation coefficient of Random Forest (RF), Metric Learning for Kernel Regression (MLKR), ChemProp, Transformer-Convolutional Neural Network (TCNN), TCNN with augmentation (TCNN Aug), Topological Regression with disjoint anchor and training set (TR*), Topological Regression (TR), and Ensemble TR on both random cross-validation and scaffold split. Notice how both TCNN and Ensemble TR (bold) achieve numerically superior performance compared to all other competing methods.

TR*, TR, and Ensemble TR) on both splitting scenarios are shown in Table 1. Figure 2 compares each method using boxplots showing the distribution of the performances for both random and scaffold splitting. As expected, TR* is unable to achieve performance comparable to the competing methods as the model is being constrained by the disjointedness requirement. When we relax this requirement, we observe that TR's predictive performance improves considerably and is only numerically inferior to TCNN with augmentation. Finally, when we incorporate an ensemble of TR models, the predictive performance of Ensemble TR is essentially as good as that of TCNN with augmentation. If we invoke the law of parsimony, our conceptually straightforward, and mathematically less complex, topological regression approach appears to be more appealing as compared to competing deep learning techniques.

**Computational comparison on ChEMBL datasets**

To illustrate the computational efficiency of TR and Ensemble TR, we report each competing method's average training time, testing time, and peak RAM consumption across all 530 datasets. These results are shown in Table 2. For fair comparison and to provide the best optimized hardware for each model, we trained the deep learning models on systems with GPUs as the training of deep learning-based models are better optimized in GPU based systems. Since the pre-trained TCNN model was released and used for fine-tuning, the reported TCNN time does not include pre-training time. From the results, we observe that TR and Ensemble TR result in the fastest training times and significantly less peak RAM consumption. For testing, TR takes more time than MLKR since RBF kernels are employed compared to MLKR which simply uses 5-NN predictions after transformation, however TR still results in faster test times than TCNN. These results demonstrate the computational efficiency of TR.

**Interpreting TR**

Inspection of the regression coefficients in $B$ demonstrates how TR offers more flexibility as compared to standard KNN. Recall, $W_{K,m}, K \in I^*, m \in I$ quantifies the impact of $Y_K$ on $Y_m$. Now, in an ordinary KNN inverse distance weighting scheme, as distance between the $K$th instance and $m$th instance increases in the chemical space, $W_{K,m}$ decreases, i.e., $\frac{\delta}{\delta d^2_{K,m;X}} W_{K,m} < 0$. However, for TR $\frac{\delta}{\delta d^2_{K,m;X}} W_{K,m} = W_{K,m}b_{KK}$. Now $W_{K,m} > 0$ by construction, therefore $sign(\frac{\delta}{\delta d^2_{K,m;X}} W_{K,m})$ depends upon the $sign(b_{KK})$. Hence, TR can push molecules closer in chemical space far apart in the response space. What this implies is, the

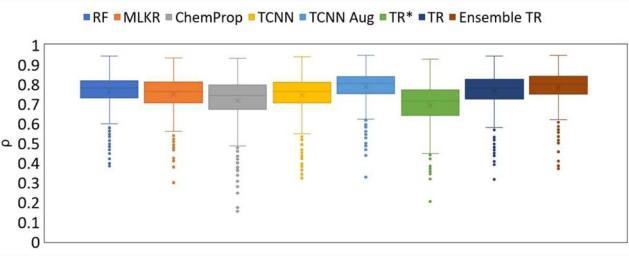

(a) CV Split Spearman's ρ

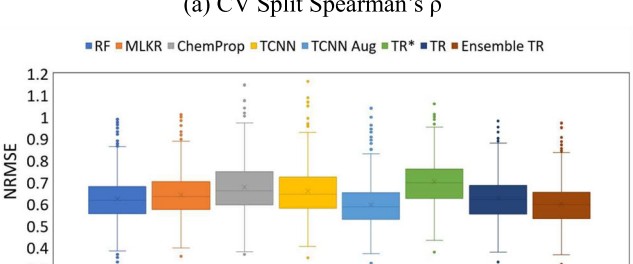

(b) CV Split NRMSE

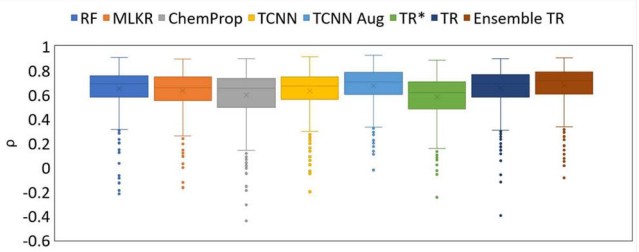

(c) Scaffold Split Spearman's ρ

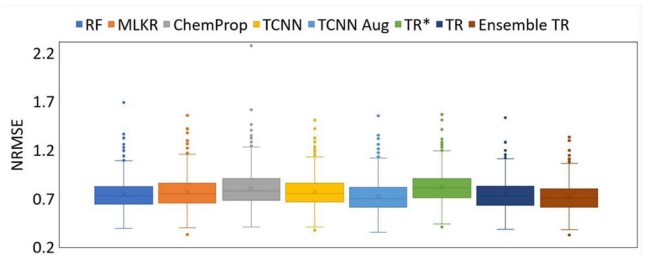

(d) Scaffold Split NRMSE

**Fig. 2 | Comparative analysis of model performances on the 530 ChEMBL bioactivity datasets. a** Average 5-fold Cross-Validation (CV) Spearman's correlation coefficient ($\rho$), (**b**) average 5-fold CV Normalized Root Mean Square Error (NRMSE), (**c**) Scaffold split Spearman's $\rho$, and (**d**) Scaffold split NRMSE. The experiment was performed on $n = 530$ ChEMBL bioactivity datasets with Random Forest (RF), Metric Learning for Kernel Regression (MLKR), ChemProp, Transformer-Convolutional Neural Network (TCNN), TCNN with augmentation (TCNN Aug), Topological Regression with disjoint anchor and training set (TR*), Topological Regression (TR), and Ensemble TR on both random cross-validation and scaffold split. The box plots show the median (central line), the interquartile range (upper and lower limits of the box), and the 5% and 95% limits (whiskers), as well as the outliers. Source data are provided in the Source Data file.

prediction generation process for TR can be interpreted in the same vein as that used by KNN, except, unlike KNN, TR searches for nearest set of anchor points in the response space.

We use the chemical space of the drugs targeting Phospholipase D2 (ChEMBL ID: CHEMBL2734) to demonstrate this phenomenon. In Fig. 3 we seek to predict the response corresponding to the molecule

**Table 2 | Computational complexity comparison of competing methods showing training time, testing time, and peak RAM consumption on the scaffold split**

| Method | CPU System (AMD EPYC 7702, 2.0 GHz, 64 Physical Cores, 128 Logical Cores, 512 GB RAM) | | | GPU System (Intel Xeon Gold 6242, 2.8 GHz, 16 Physical Cores, 32 Logical Cores, 384 GB RAM, NVIDIA Tesla V100) | | |
|---|---|---|---|---|---|---|
| | MLKR | TR | Ensemble TR | ChemProp | TCNN | TCNN Aug |
| Train Time (s) | 181.648 | 1.602 | 13.768 | 56.586 | 30.586 | 109.759 |
| Test Time (s) | 0.354 | 1.008 | 12.301 | 0.510 | 6.915 | 20.003 |
| Peak RAM (GB) | 1.777 | 0.253 | 0.314 | 2.044 | 2.864 | 3.231 |

The compared methods listed on the table are Metric Learning for Kernel Regression (MLKR), Topological Regression (TR), Ensemble TR, ChemProp, Transformer-Convolutional Neural Network (TCNN), and TCNN with Augmentation (TCNN Aug).

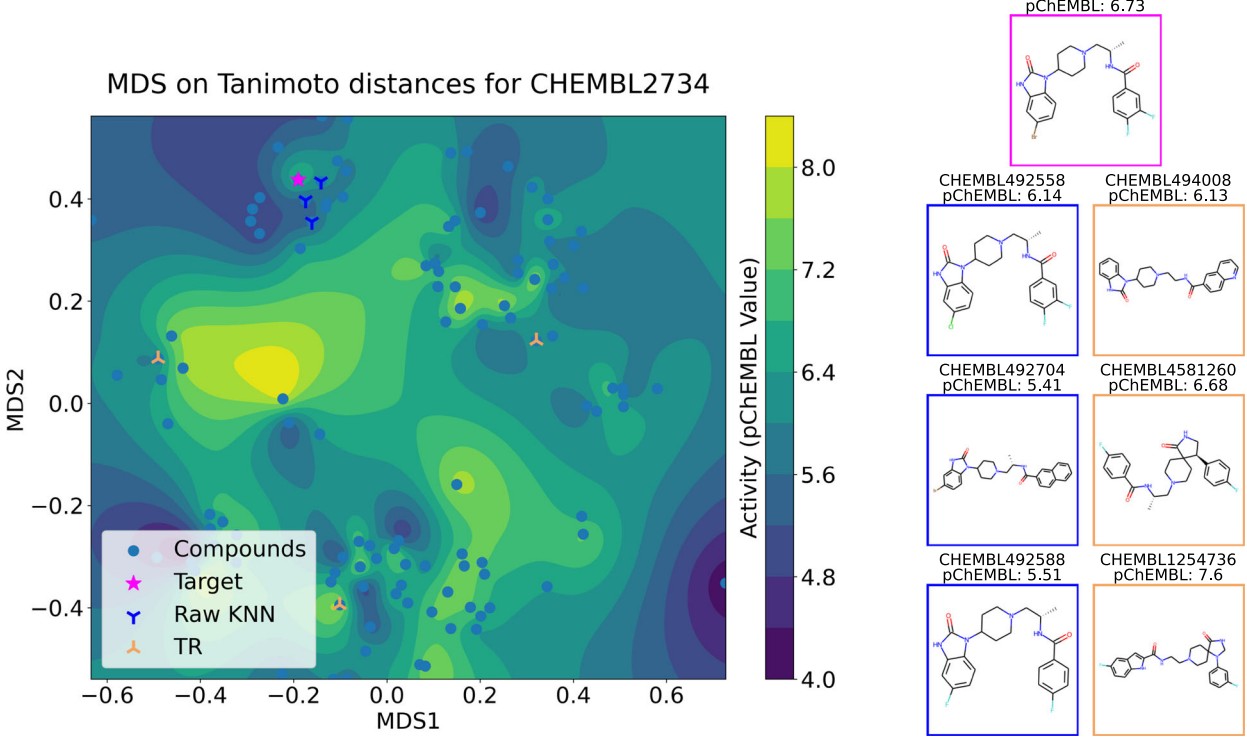

**Fig. 3 | Comparative analysis of the neighbors found by a KNN procedure and a TR procedure.** Nearest neighbors found by K-Nearest Neighbors (KNN) and Topological Regression (TR) in a single fold of the 5-fold cross-validation setup for the CHEMBL2734 dataset. KNN finds the nearest training samples and can lead to misleading results when the target (pChEMBL = 6.73) is close to an activity cliff (KNN prediction = 5.69), while TR attempts to find nearest neighbors in the response space, leading to more informed and meaningful predictions (TR prediction = 6.80). The presented chemical space was done by performing a Multidimensional Scaling (MDS) on the dataset. The presented molecules have a colored frame, where magenta is the target presented by the same color in the MDS plot, blue are the 3 nearest neighbors also presented in the MDS by the same color, and yellow is the framing the molecules found by TR, presented in the MDS by the same color.

CHEMBL492559 (denoted by a red star, pChEMBL= 6.73) in the test set. Based on similarity in the chemical space, standard KNN finds three molecules, CHEMBL492558, CHEMBL492704, and CHEMBL492588, as nearest neighbors, under a 5-fold cross-validation protocol, and makes predictions based on the average of the activities of these three molecules. However, the target molecule is almost at the edge of a high-activity region. Therefore, naive KNN identifies two neighbors, CHEMBL492704 and CHEMBL492588 from the nearby low activity region (across the cliff) and only one neighbor CHEMBL492558 from the ideal high activity region. This happens because the high-activity region in the neighborhood of the target molecule is sparsely populated. In contrast, since TR directly incorporates Y in the learning, it identifies three cross-cliff molecules, CHEMBL494008, CHEMBL4581260, and CHEMBL1254736, that have greater weights in predicting the response associated with the target molecule as compared to CHEMBL492704 and CHEMBL492588. Observe that all three

molecules identified by TR as nearest neighbors (CHEMBL494008, CHEMBL4581260, CHEMBL1254736) are in relatively high-activity regions. By presenting structures from diverse scaffolds that exhibit similar activities, TR not only enhances prediction reliability but also aids in the identification of key spatial structural characteristics influencing the activities. The presented structures can be further validated with structural chemical methods such as structural alignment or docking simulations.

To further illustrate this point across the entire dataset, rather than for one particular test molecule, we generated KNN-graphs depicting the predictions of the various similarity-based methods with the color indicating the activity elicited by the molecules. To do so, each training and test sample was represented as a node, and the predicted neighbors were considered as the connecting edges. These graphs are synonymous with NSGs, in fact, just like NSGs, the edges were only included if the similarity was greater than a fixed cutoff TC

and if the molecules were predicted as one of the nearest neighbors. Therefore, the number of neighbors and the cutoff TC control the connectedness of the network graphs, more connections would be established with a larger number of nearest neighbors and lower cutoff similarities until the graph is complete. We used 5 nearest neighbors and the mean similarity of the entire target dataset as the cutoff TC for each competing method for all subsequent network graphs, meaning at most 5 connections would be established if their similarities were greater than the fixed cutoff TC. An example of these KNN-graphs, depicting the test nearest neighbors of a single CV fold of the dataset CHEMBL2734, is included in Fig. 4. Additional figures depicting the training predictions, testing predictions, and molecules within the most active cluster are included in the supplementary document. Notice that the predicted TR neighbors are similar in response value, leading to more homogeneous activity throughout the clusters, whereas KNN and MLKR both result in clusters containing diverse activity values. To quantify this variability, we included the average within-cluster standard deviations for each method in the figure where a low within-cluster standard deviation denotes a more homogeneous cluster.

To systematically show this behavior across all 530 datasets, we calculated the average within-cluster standard deviation from the foregoing test prediction KNN-graphs for the competing methods. Figure 5 depicts these results in the form of a line graph across all 530 datasets. Clearly, TR systematically produces lower within-cluster

standard deviation compared to KNN and MLKR, resulting in higher levels of homogeneous activity within the clusters. If we envision activity cliffs to be a phenomenon that induces a strong outlier within an otherwise homogeneous cluster, then it stands to reason that by measuring within-cluster homogeneity we can infer about the presence of cliffs in that cluster. Higher levels of within-cluster homogeneity essentially smooths out activity cliffs resulting in more relevant similarity-based predictions and providing practitioners with instance-wise similar molecules for lead optimization.

Since TR results in more homogeneous clusters, the clusters themselves can be more meaningfully mined by chemists for innovative design ideas, potential target leads, and lead optimization pathways. For example, clustering can be performed on the training data, and the most active cluster may contain molecules with specific features that practitioners can use to guide designs and future experiments. The same can be done with the least active cluster to see which molecular features to avoid and provide further insights. Furthermore, the most active training cluster can be mined for lead molecules that have other desired characteristics, such as low toxicity or ease of production. Analogous to NSGs, the training clusters can also be used to visualize lead optimization pathways. Figure 6 depicts a lead optimization pathway in the most active cluster of target protein complex Integrin alpha-4/beta-7 (CHEMBL278) with (a) the TR KNN-graph obtained from the training data of a single CV fold, (b) the most active cluster depicted as a minimum spanning tree with the minimum

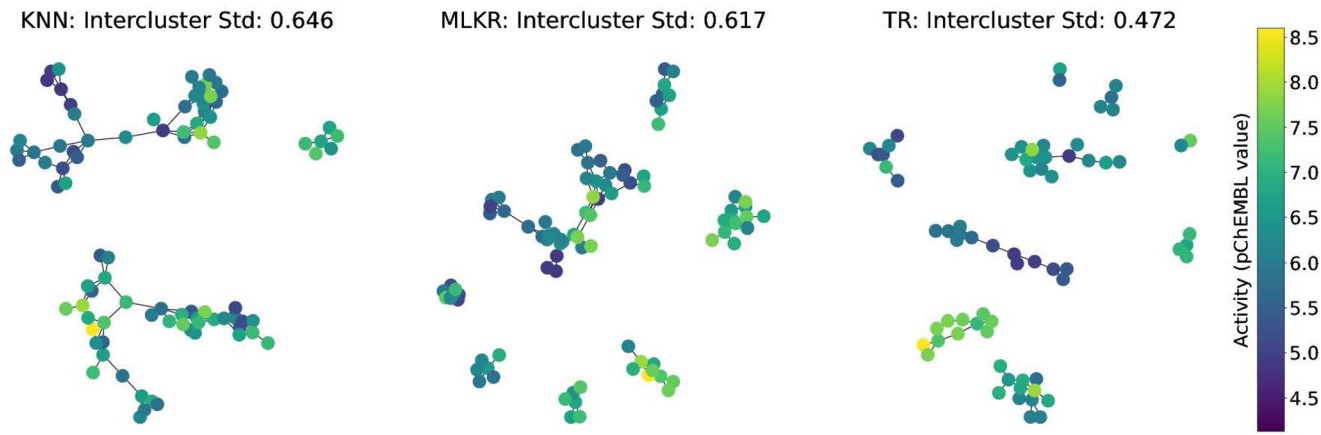

**Fig. 4 | k-Nearest neighbor graphs visualizing the 5 nearest neighbor predictions of K-nearest neighbors (KNN), metric learning for kernel regression (MLKR), and topological regression (TR).** For this experiment, we used the foregoing target activity Phospholipase D2 (CHEMBL2734), and computed the 5-nearest neighbors for the three different methods, using the mean similarity of the dataset as the neighbor similarity cutoff. We can clearly see that the intercluster standard deviation is minimized by using the TR procedure.

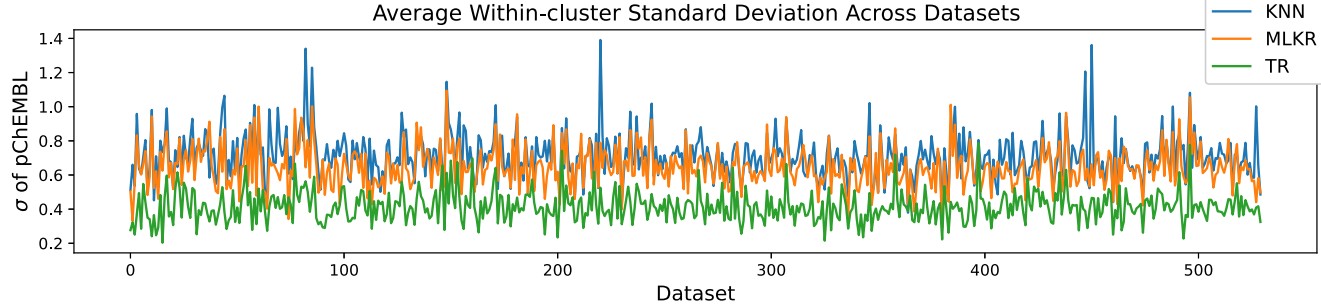

**Fig. 5 | Intercluster standard deviation computed accross the 530 ChEMBL datasets.** For this experiment, we show a quantitative comparison across all 530 datasets showing the average within-cluster standard deviation ($\sigma$) of pChEMBL values obtained from the test prediction KNN-graphs for K-Nearest Neighbors (KNN), Metric Learning for Kernel Regression (MLKR), and Topological Regression (TR). It can be seen that the average standard deviation is consistently lower for TR compared to the baseline methods. Source data are provided in the Source Data file.

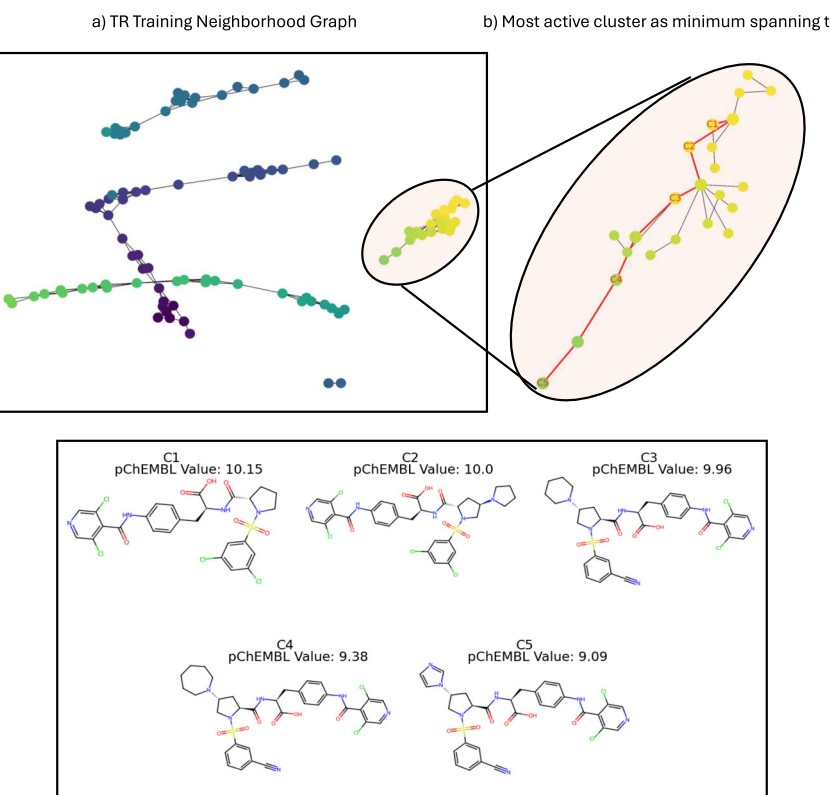

**Fig. 6 | Optimization pathway visualization in the most active training cluster of target protein complex Integrin alpha-4/beta-7 (CHEMBL278). a** depicts the training neighborhood graph obtained from Topological Regression (TR)

predictions, (**b**) depicts the minimum spanning tree of the most active cluster with a minimum path connecting the most active and least active molecules in red, and (**c**) depicts 5 example molecules showing the lead optimization pathway.

spanning path between the most active and least active molecules depicted in red, and (c) 5 example molecules from the lead optimization pathway connecting the most active and least active molecules in the cluster. These pathways can be traversed by chemists to envision what changes resulted in specific behaviors, allowing them to easily analyze the current state of a target dataset and discover potential design ideas (additional figures representing optimization pathways for various target datasets are provided in the Supplementary document). If we envision an untested molecule as an additional node in Fig. 6a, the TR method could directly produce the set of edges radiating from that node (via the model for W) that would enable one to assess how the untested molecule relates with the previously tested molecules. This could enable greater trust in the predictions as the chemist could easily visualize how the new sample relates to known molecules. Additionally, these graphs fit directly with Laplacian Scores for feature selection, allowing global feature importance to be calculated in a routine fashion. Lastly, when paired with SHAP or MMACE, which are model agnostic, TR would be able to efficiently generate instance-wise feature importance and unseen counterfactuals, adding additional layers to TR's interpretability.

## Discussion

In this paper, we have developed a statistical methodology, topological regression (TR), to perform similarity-based regression and demonstrated how it can be used for QSAR modeling. We tested TR on regression tasks with 530 ChEMBL human targets and compared it with a traditional RF, Nearest Neighbors, a metric learning algorithm (MLKR), and two deep learning methods, ChemProp and Transformer-CNN. Empirically, we observed that TR or ensemble TR compared favorably against all competing methods in terms of predictive accuracy on the scaffold split and achieved comparable performance with

TCNN on the random splitting at a much lower computational cost. Most importantly, TR provides explainability, visual interpretability, and theoretical justifiability in the form of testable adequacy and optimal model size.

The performances of RF, TCNN, ChemProp, and MLKR are mostly interpreted in a comparative sense. The usual measures employed to assess the performance of these models - NRMSE, MAE - have unbounded support and hence do not offer information about the goodness-of-fit. TR on the other hand completely relies on multivariate general linear models - geographically weighted regression when extracting $W_{i,j}$ from the drug response, and standard regression theory when modeling $W_{i,j}$. For both of these techniques, rigorous tests for goodness-of-fit exist[56,57]. Since the standard coefficient of determination offers an immediate goodness-of-fit statistic for linear models (or transformed linear models), we compute the training R-sq values (using (7)) for all 530 ChEMBL datasets considered in this paper. The average R-sq turns out to be 0.8396. Evidently, our conceptually straightforward parametric linear model has sufficient power to explain variation in $W_{i,j}$. Turning to predictive adequacy, we compute the prediction interval for the W's (using extracted W's as targets) in the cross-validation set. Once again, the linear model specification allows us to compute the prediction interval analytically. We then compute the coverage of these prediction intervals across all folds. Ideally, we would like to see the coverage of the prediction interval achieve a nominal level. In all the 530 datasets across all the folds, the coverage of 95% prediction interval is 94.3%. Clearly, the model specified in (7) is adequate for prediction purposes as well. These results provide empirical justification for the adequacy of the TR model.

Given the small to moderate sample size in ChEMBL datasets, model complexity has a significant impact on prediction performance. For ChemProp or TCNN-type deep learners, regularization of network

weights, drop-out layers, and ablations are standard procedures to control model complexity. However, these measures are adhoc and their theoretical properties are not well established. For standard KNN (or even in MLKR), the number of neighbors determines the model size. However, we need to fix the number of nearest neighbors a-priori and tune that quantity via cross-validation. TR, on the other hand, offers a theoretically appropriate way to choose neighborhood size and hence model complexity. In TR, the anchor points play the role of neighbors and $|I^*|$ determines the size of the coefficient matrix $B$. Consequently, changing $|I^*|$ yields sequences of nested models, and hence standard model selection techniques, for instance, AIC or BIC, could be used to identify the appropriate size of $I^*$ without resorting to cross-validation. Since AIC/BIC automatically penalizes model complexity for a given sample size, we can arrive at an optimal model complexity for TR.

Furthermore, TR provides an intuitive explanation of its predictive mechanism based on nearest neighbors in the response space as shown through KNN graphs in the section "interpreting TR". This explanation could be gleaned from MLKR as well. However, the computational complexity associated with semi-definite programming, required in MLKR, is considerable if the dimension of the input space is high. TR, on the other hand, directly learns the weights associated with neighboring responses, and, by a suitable transformation, estimates the parameters in an unconstrained fashion. This leads to a significant reduction in computational expense as reported in Table 2.

Finally, the visual representation of TR's predictive mechanism could provide design ideas and allow fast knowledge-based model validation. We anticipate that our framework will have practical value in drug discovery or other QSAR tasks and assist in designing new molecules more effectively.

## Methods
### Data description and problem motivation
We begin with a description of the datasets that we use to illustrate the comparative performances of the competing models. We offer a brief description of ChemProp, Transformer-CNN and MLKR methods and then outline the motivation behind developing the TR framework.

**Dataset.** Since our focus is on QSAR modeling in the lead optimization phase of drug discovery, we choose to assess the performance of competing models on well-curated datasets with single target bioactivity. For this purpose, we downloaded data from the ChEMBL database[58] following the extraction protocol of[59]. This included only selecting 'SINGLE PROTEIN' or 'PROTEIN COMPLEX' human targets with confidence scores of 9 and 7, respectively. Additionally, only pCHEMBL values, which are comparable bioactivity measures of half-maximal response (IC50, XC50, EC50, etc.) on a negative logarithmic scale, were selected. We refer the readers to[59] for further data extraction details.

In the cleaning phase, we first removed the datasets that were too small to train ChemProp and Transformer-CNN. Within each dataset, we further removed instances with duplicated SMILES and instances with chemically invalid SMILES strings which could not be converted to RDKit molecules. Finally, we had 530 datasets on various human target bio-activities. Sample size ranged from 100 to 7890 with the median sample size being 677. The various target activities, referred to as pChEMBL values, were used as the univariate response variable.

Although several representative descriptors and fingerprints (for example: RDKit descriptors, Mordred[6], ECFP4[7]) are available, we mainly focus on ECFP4 representation for similarity-based predictive models because, empirically, this representation offered the best predictive performance. We relegate the results demonstrating the superior predictive performance of the ECFP4 representation to the Supplementary Material. We calculate folded ECFP4 fingerprints using RDKit's implementation of the Morgan algorithm with a radius of

2 atoms and bit-size of 1024. Since the output of this representation system is binary, we use the Tanimoto coefficient (TC) as a measure of similarity and $1 - TC$ as a measure of distance for TR. No standardization steps were required as RDKit was used to extract ECFP4 fingerprints. The ECFP4 fingerprints were used to train the RF model, whereas Chemprop used the SMILES string inputs to internally extract the graph representations and Transformer-CNN directly used the SMILES strings.

**ChemProp.** We used ChemProp as a baseline model because of its demonstrated utility in drug discovery. ChemProp is a full-fledged Graph Convolutional Neural Network model that takes 2D representations of molecules as predictors. We employed ChemProp's Bayesian hyperparameter optimization, which optimizes the hidden size, depth, dropout, and the number of feed-forward layers, and trained the model for 100 epochs for all datasets.

**Transformer-CNN.** We also used Transformer-CNN (TCNN) as a baseline model as it is self-proclaimed to be a Swiss-army knife for QSAR modeling. TCNN is a pre-trained model on over 17 million pairs of strings for the task of SMILES canonicalization. The output of the transformer encoder is then used to generate model-acquired FPs, which are used for downstream prediction through task-trained Text-CNN and convolutional highway layers. In addition, the architecture enables data augmentation by ensembling the results from multiple non-canonical smiles for each sample. Lastly, the architecture contains practically no hyperparameters and enables learning rate scheduling and early stopping, limiting the need for hyperparameter optimization. This mixture of large pre-training, sample augmentation, and string-size agnostic architecture results in a powerful prediction model. We followed the TCNN instructions and trained the model on the SMILES strings, with and without augmentation, for at most 35 epochs as learning rate scheduling and early stopping were employed.

**Metric Learning Kernel Regression.** The purpose of metric learning is to find a distance metric for a specific task through supervised learning. The metric found by metric learning could subsequently be used in KNN regression or kernel regression for generating predictions and visualizations. For regression tasks, MLKR[51] finds the Mahalanobis metric that minimizes the cumulative leave-one-out CV error $\mathcal{L} = \sum_i (Y_i - \hat{Y}_i)^2$, where $Y_i$ is the numeric response variable of the i-th training sample and $\hat{Y} = \frac{\sum_{j \neq i} Y_j W_{ij}}{\sum_{j \neq i} W_{ij}}$ with $W_{...}$ being the weights associated with Gaussian kernels. In particular, the transformation matrix $L$ used to obtain the learned metric can be written as a decomposition of Mahalanobis matrix $M = L^T L$. After $L$ is learned from the data, the original coordinate system of the predictor space $\boldsymbol{X}$ is transformed into the new coordinate system given by $\boldsymbol{LX}$. Thus, MLKR learns a global space transformation, which can be used to calculate the distance in the response space. Then KNN-regression or similarity-based kernel regression can be performed to provide predictions and interpretation.

However, in order to compute distances, we first need to characterize the molecules in a fashion such that distances can be computed. As mentioned, we focus on ECFP4 fingerprints, which is thus the initial coordinate system supplied to MLKR to learn the transformation and produce a new coordinate system such that the predictor space is approximately isometric to the response space. Figure 7 illustrates this phenomenon. In the left panel, we computed the pairwise Tanimoto distances among all the molecules targeting Mitogen-Activated Protein Kinase 12 (ChEMBL ID: CHEMBLE1908389) using ECFP4 features and projected them in 2D MDS space. The intensity of the pixels indicate the response each molecule elicited. In the right panel, we used the distance metric learned from MLKR to generate the 2D coordinates. Observe how the two molecules, CHEMBL3727733 and CHEMBL3729567, which appeared to be neighbors in the chemical

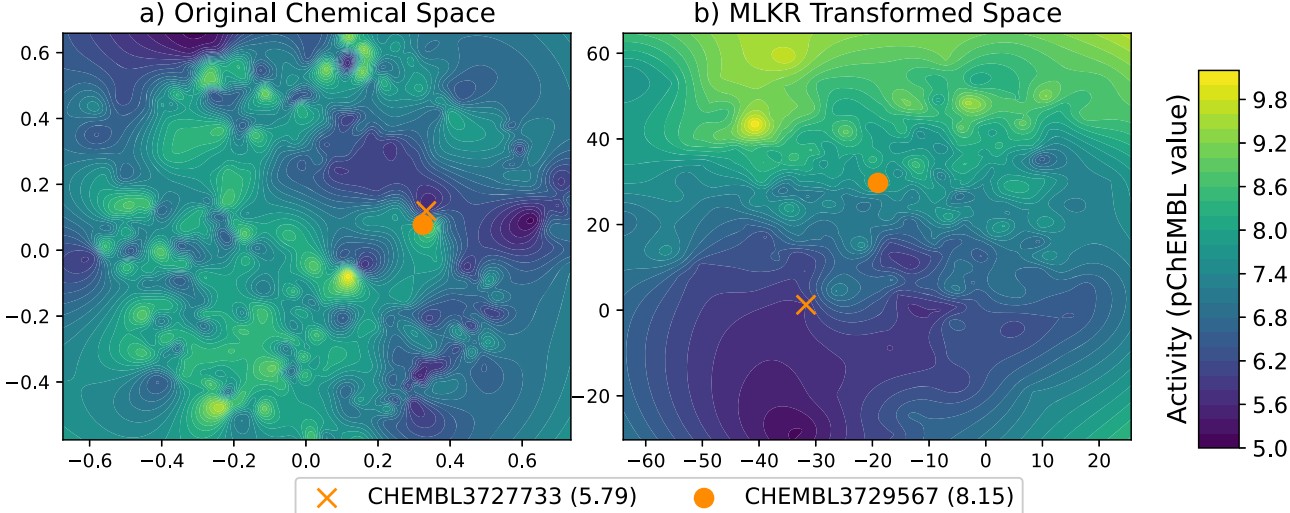

**Fig. 7 | The 2-D multidimensional scaling (MDS) of the original chemical space and metric learning for kernel regression (MLKR) transformed space of ChEMBL target mitogen-activated protein kinase 12 (CHEMBL1908389).** The transformed and interpolated chemical space with (**a**) MDS using Tanimoto distance and (**b**) MLKR. Notice how MLKR smooths the activity space and separates the two similar molecules CHEMBL3727733 (pChEMBL = 5.79) and CHEMBL3729567 (pChEMBL = 8.15) compared to the original MDS transformed space.

space, were pushed apart after the MLKR transformation. Additionally, we observe a smoother spatial trend in the image produced after the MLKR transformation which allows us to use KNN or kernel regression - with purely distance-dependent kernel elements - for prediction purposes.

**Comparison procedure.** To compare model performances we design two types of data splits: (**a**) random split and (**b**) scaffold split. Random split is done with 5-fold cross-validation with 80% training and 20% testing in each fold. The scores of the five folds are averaged as the final score. In drug discovery, new structures are often proposed by editing on the scaffold of a known good candidate. Predictions are more likely to fail across scaffolds due to greater chemical dissimilarities. Scaffold split makes sure the training and test samples belong to different Murcko scaffolds - mimicking scenarios when predictions for a new structure of a different scaffold is sought. Since full-blown cross-validation is not feasible with scaffold splits, we use a single hold-out set comprising approximately 20% of data points for each ChEMBL dataset. We use Spearman $\rho$ and Normalized Root Mean Square Error (NRMSE) to compare the candidate models' capabilities to generate predictions. In the section "Results", we compare these two metrics obtained from ChemProp with those obtained from MLKR-KNN under both splitting scenarios for all 530 ChEMBL datasets and observe that MLKR-KNN offers numerically superior performance as compared to ChemProp, even though MLKR is not directly a regression technique.

This empirical observation motivates us to develop TR based on a distance formulation and thereby make the MLKR-type strategy amenable to statistical inference. We observe that in the MLKR procedure, a lot of effort is undertaken to ensure that the transformed space is indeed a metric space. However, for prediction, a weighted averaging of the responses from nearest neighbors is performed. Notice that symmetry and non-negativity are the only two conditions required for those weights ($W_{ij}$). Therefore, we contend that we can directly work with $W_{ij}$s instead. We then proceed to show that, under suitable distributional specification, an explicit estimator of $E(W_{ij})$ could be obtained. Since the estimand is an expectation operator, standard statistical theory (delta method, residual bootstrap) could be brought to bear to assess the statistical properties of this estimator. To the best of our knowledge, such statistical assessment of the estimates produced by vanilla MLKR is not available.

## Multivariate construction of topological regression

Topological regression (TR) is a similarity-based regression framework that connects the distances in the chemical space with non-negative weights appearing in nearest neighbor regression defined on the response space. The model is illustrated in Fig. 8. More specifically, we specify a multivariate regression model for the weights $W_{ij}$s and derive a closed-form expression for the estimator of $E(W_{ij})$ under an inverse distance-weighting scheme. Subsequently, we also offer a discussion on an approximate estimator of the foregoing quantity when the weighting is done using a Gaussian kernel.

Let $\mathcal{D}$ represent the set of all training points. First, we partition $\mathcal{D}$ into a set of $K$ anchor points and $N = |\mathcal{D}| - K$ neighborhood-training points. Let $I^* = \{i_1^*, i_2^*, \ldots, i_K^*\}$ be the set of indices associated with the anchor points and $I = \{i_1, i_2, \ldots, i_N\}$ be the indices associated with the neighborhood-training points, with $I^* \cap I = \phi$ and $|I^*| < |I|$. Let $Y_{i_j}, i_j \in I$ be the response associated with the $i_j$th instance in the set $I$. Our goal is to express $Y_{i_j}$ as a linear combination of responses $Y_{i_l^*}$ belonging to the set $I^*$, i.e.

$$Y_{i_j} = \sum_{i_l^* \in I^*} W_{i_l^* i_j} Y_{i_l^*}, \forall i_j \in I \qquad (1)$$

where $W_{i_l^* i_j}$ is a non-negative weight that determines the contribution of the response associated with the $l$th point in $I^*$ towards the response associated with the $j$th point in $I$. Such non-negative weights are fairly common in distance-weighted regression, for instance, in geographically weighted spatial regression models, often the weights are specified in terms of Gaussian kernels, i.e., $W_{i_l^* i_j} = \exp(-\beta d_{i_l^*, i_j}^2)$ with $d^2(.)$ being a squared Euclidean distance and $\beta > 0$ controlling the smoothness of the random field.

**Neighborhood training model.** Customarily, the weights are expressed as a deterministic function of the distances in the predictor space. In standard KNN regression, we assume that distance in the predictor space is proportional to the distance in the response space. In metric learning, a transformation of the predictor space is learned such that there is an approximate isometry between the transformed predictor space and the response space. In TR, we instead write a formal

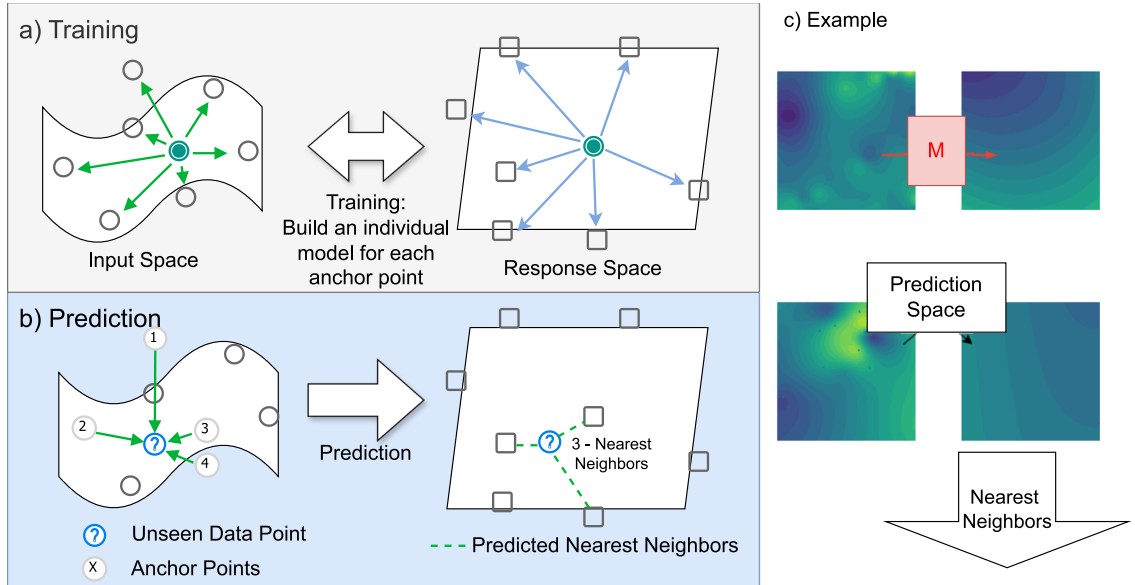

**Fig. 8 | Overview of the proposed topological regression framework. a** Distances in the input space are used to predict distances in the response space, which can subsequently be paired with similarity-based prediction methods such as nearest neighbor or kernel regression. **b** Targets of the unseen predictions will be calculated using the anchor points, and these results can be compared easily with other similarity-based methods, like the K-Nearest Neighbors (KNN). **c**) The whole procedure is shown in an example, where we can see the predicted space by the model (M) after training.

statistical model to connect $W_{i_j^* i_j}$ with the squared Euclidean distances in the predictor space in the following fashion:

We define the weights

$$W_{i_j^* i_j} = 0 \text{ if } i_j^* = i_j$$
$$> 0 \text{ if } i_j^* \neq i_j \tag{2}$$

and since we have $I^*$ and $I$ to be disjoint and the responses could be assumed to be absolutely continuous, we can define

$$\tilde{W}^{N \times K} = \begin{bmatrix} \log(W_{i_1^*, i_1}) & \log(W_{i_2^*, i_1}) & \cdots & \log(W_{i_K^*, i_1}) \\ \log(W_{i_1^*, i_2}) & \log(W_{i_2^*, i_2}) & \cdots & \log(W_{i_K^*, i_2}) \\ \cdots & \cdots & \cdots & \cdots \\ \log(W_{i_1^*, i_N}) & \log(W_{i_2^*, i_N}) & \cdots & \log(W_{i_K^*, i_N}) \end{bmatrix} \tag{3}$$

with the entries in $\tilde{W}$, i.e., $(\tilde{W})_{i_j^* i_j}$ being real quantities. Define the squared Euclidean distance matrix in the predictor space as

$$D_X^{N \times K} = \begin{bmatrix} d_{i_1^*, i_1;X}^2 & d_{i_2^*, i_1;X}^2 & \cdots & d_{i_K^*, i_1;X}^2 \\ d_{i_1^*, i_2;X}^2 & d_{i_2^*, i_2;X}^2 & \cdots & d_{i_K^*, i_2;X}^2 \\ \cdots & \cdots & \cdots & \cdots \\ d_{i_1^*, i_N;X}^2 & d_{i_2^*, i_N;X}^2 & \cdots & d_{i_K^*, i_N;X}^2 \end{bmatrix} \tag{4}$$

We define a simple multivariate linear regression model connecting $\tilde{W}$ with $D_X$. Consider the $m$th row of $\tilde{W}$. Observe that, this row consists of the weights used to express the $m$th response in $I$ using all the responses in $I^*$. We envision this row to be a set of repeated measurements taken on the $m$th point in $I$ from the *vantage points* in $I^*$. Thus, denoting the $K$ elements in the $m$th row of $\tilde{W}$ by $\tilde{W}_{.,m} = (\tilde{W}_{1,m}, \tilde{W}_{2,m}, \cdots, \tilde{W}_{K,m})$, the corresponding row of predictors in $D_X$ by $D_{.,m;X} = (d_{1,m;X}^2, d_{2,m;X}^2, \cdots, d_{K,m;X}^2)$, and the matrix of regression

coefficients by

$$B^{K+1 \times K+1} = \begin{bmatrix} b_{01} & b_{02} & \cdots & b_{0K} \\ b_{11} & b_{12} & \cdots & b_{1K} \\ \cdots & \cdots & \cdots & \cdots \\ b_{K1} & b_{K2} & \cdots & b_{KK} \end{bmatrix} \tag{5}$$

we arrive at the following regression model

$$\begin{aligned}
\tilde{W}_{1,m} &= b_{01} + b_{11} d_{1,m;X}^2 + b_{21} d_{2,m;X}^2 + \cdots + b_{K1} d_{K,m;X}^2 + \epsilon_1 \\
\tilde{W}_{2,m} &= b_{02} + b_{12} d_{1,m;X}^2 + b_{22} d_{2,m;X}^2 + \cdots + b_{K2} d_{K,m;X}^2 + \epsilon_2 r \\
&\cdots \\
\tilde{W}_{K,m} &= b_{0K} + b_{1K} d_{1,m;X}^2 + b_{2K} d_{2,m;X}^2 + \cdots + b_{KK} d_{K,m;X}^2 + \epsilon_K
\end{aligned} \tag{6}$$

with $\boldsymbol{\epsilon} = (\epsilon_1, \epsilon_2, \cdots, \epsilon_K) \sim \mathcal{N}_K(0, \Sigma)$. Now assuming mutual independence across the $N$ rows of $\tilde{W}$ and since $N > K$ (by construction), we can obtain the MLEs of $B$ and $\Sigma$. Let $\hat{B}$ and $\hat{\Sigma}$ denote their respective estimates. Then, for a new query point, we can compute $(d_{1,query;X}^2, d_{2,query;X}^2, \cdots, d_{K,query;X}^2)$ and, using $\hat{B}$, obtain the predictions $(\tilde{W}_{1,query}, \tilde{W}_{2,query}, \cdots, \tilde{W}_{K,query})$. However, observe that (1) requires $(W_{1,query}, W_{2,query}, \cdots, W_{K,query})$ to generate a prediction for the query point, and simply exponentiating the output, $\hat{\tilde{W}}$, of (6) will yield a biased estimate of $W$ because $E(W) = E(e^{\tilde{W}}) \neq e^{E(\tilde{W})}$ due to Jensen's inequality. Therefore, we use the properties of the multivariate log-normal distribution to improve the estimate of $W$ in the following way:

Clearly $\boldsymbol{W}_{.,m} = e^{\tilde{\boldsymbol{W}}_{.,m}}$ where the exponent is taken coordinate-wise with $\tilde{\boldsymbol{W}}_{.,m} \sim \mathcal{N}_K(\boldsymbol{\mu}_{.,m}, \Sigma)$ and $\mu_{j,m} = b_{0j} + b_{1j} d_{1,m}^2 + b_{2j} d_{2,m}^2 + \cdots + b_{Kj} d_{K,m}^2$. Then the usual relationship between the expectation of a log-normal variate with the moment-generating function of its normal counterpart can be used to show that $E(W_{j,m}) = E(e^{\tilde{W}_{j,m}}) = \exp(\mu_{j,m} + \Sigma_{jj}/2)$. Additionally, it is fairly straightforward to show that the covariance matrix of $\boldsymbol{W}_{.,m}$ is given by $Var(\boldsymbol{W}_{.,m}) = diag(E(\boldsymbol{W}_{.,m}))(e^{\Sigma} - \boldsymbol{1}\boldsymbol{1}^T)diag(E(\boldsymbol{W}_{.,m}))$. Consequently, an estimator of $\boldsymbol{W}_{j,query}$ is given by $\hat{W}_{j,m^*} = \hat{E}(\boldsymbol{W}_{j,query}) = \exp(\hat{\mu}_{j,query} + \hat{\Sigma}_{jj}/2)$ and the corresponding estimator of the covariance matrix is $\hat{Var}(\boldsymbol{W}_{.,query}) = diag(\hat{\boldsymbol{W}}_{.,query})(e^{\hat{\Sigma}} - \boldsymbol{1}\boldsymbol{1}^T)diag(\hat{\boldsymbol{W}}_{.,query})$. The estimated

covariance matrix is positive definite as long as $\hat{\Sigma}$ is positive definite. Furthermore, since $\hat{B}$ is asymptotically normally distributed, we can obtain a conservative estimate of the pointwise prediction interval of $W_{\cdot,m\cdot}$ using the parametric bootstrap technique outlined in[60].

**Extraction of W.** In the above discussion, we have used $\log(W)$ as the target of the multivariate regression in (6). However, $W$ are not observed, but are parameters that appear in the distance-weighted regression in the response space (1). Hence, we first need to extract these weights. A naive option is to set the weights $W_{i_j^*,i_j}$ as the inverse of squared Euclidean distance in the response space between points in $I$ and $I^*$, i.e. $W_{i_j^*,i_j} = 1/d^2_{i_j^*,i_j;Y}, i_j^* \in I^*, i_j \in I$. In this configuration, we can simply supply $1/d^2_{i_j^*,i_j;Y}$ in the LHS of (6). We will still recover a closed form expression for $\hat{E}(W)$ because the log-normal distribution is closed under an inverse transformation.

### Univariate construction of topological regression

The requirement $I^* \cap I = \phi$ in the previous section induces a delicate trade-off. If we increase the number of anchor points, the neighborhood training model becomes overparametrized. If, on the other hand, we decrease the number of anchor points there may not be enough anchor points to reliably estimate the response, especially in isolated regions of high activity.

One possible solution is to bring the distances among anchor points themselves in the neighborhood training model. But, that conflicts with the above theoretical development because each point in $I^*$ can be observed from the remaining $K-1$ points in $I^*$ and hence we do not have a $K \times K$ covariance matrix. Additionally, because of the symmetry constraint ($W_{i,j} = W_{j,i}$), we can only work with the triangular matrix of weights associated with points within $I^*$. Thus, if we forego the above multivariate log-linear regression construction (6) and view TR purely as a least-square optimization problem we can use $K(N-K) + K(K-1)/2$ equations to obtain the least-square estimates of the coefficient matrix $B$. In this scenario, the first $K(N-K)$ equations are obtained by varying $m$ from $1, 2, \ldots N$ in (6). The remaining $K(K-1)/2$ equations connect the $\tilde{W}_{i_j^*,i_j}$ with the instances in $I^*$. More specifically, dropping the subscript $i$ and simply denote the $K$ elements in $I^*$ as $\{1^*, 2^*, 3^*, \cdots, K^*\}$, then we have the following system of equations:

$$
\begin{aligned}
\tilde{W}_{2^*,1^*} &= b_{02} + b_{12}d^2_{1^*,1^*;X} + b_{22}d^2_{2^*,1^*;X} + \cdots + b_{K2}d^2_{K^*,1^*;X} + \epsilon_{2^*1^*} \\
\tilde{W}_{3^*,1^*} &= b_{03} + b_{13}d^2_{1^*,1^*;X} + b_{23}d^2_{2^*,1^*;X} + \cdots + b_{K3}d^2_{K^*,1^*;X} + \epsilon_{3^*1^*} \\
&\cdots \\
\tilde{W}_{3^*,2^*} &= b_{03} + b_{13}d^2_{1^*,2^*;X} + b_{23}d^2_{2^*,2^*;X} + b_{33}d^2_{3^*,2^*;X} + b_{K3}d^2_{K^*,2^*;X} \cdots + \epsilon_{3^*2^*} \\
&\cdots \\
\tilde{W}_{K^*,(K-1)^*} &= b_{0K} + b_{1K}d^2_{1^*,(K-1)^*;X} + b_{2K}d^2_{2^*,(K-1)^*;X} + b_{3K}d^2_{3^*,(K-1)^*;X} + \cdots \\
&\quad + b_{KK}d^2_{K^*,(K-1)^*;X} + \epsilon_{K^*(K-1)^*}
\end{aligned}
$$

(7)

$\hat{B}$ could be obtained by minimizing the error sum of squares. Additionally, if we assume the error terms are iid $\mathcal{N}(0,\sigma^2)$, we can easily obtain $\hat{\sigma}^2$ from the residuals. Now, when a query instance comes in with known chemical features, we can compute $\boldsymbol{d}^2_{\cdot,query} = [d^2_{1^*,query}, d^2_{2^*,query}, \cdots, d^2_{K^*,query}]$ in the chemical space and obtain $\hat{\tilde{\boldsymbol{W}}}_{\cdot,query} = \boldsymbol{d}^2_{\cdot,query}\hat{B}$. Then an estimator of the neighborhood weights for the query point is given by $\hat{\boldsymbol{W}}_{\cdot,query} = \exp(\hat{\tilde{\boldsymbol{W}}}_{\cdot,query} + \hat{\sigma}^2/2)$.

Additionally, since the $W$'s in this case are univariate, we have the flexibility to write $W_{i_j^*,\neg i_j^*} = \exp(-\beta d^2_{i_j^*,\neg i_j^*;Y})$ with $\beta > 0$ and replace the

$W$'s in the LHS of (7) by $\log(d^2_{i_j^*,\neg i_j^*;Y})$. Now, each $d^2$ has a univariate lognormal distribution. Now, to obtain an estimator of $E(W_{i_j^*,\neg i_j^*})$, we first observe that

$$
E(W_{i_j^*,\neg i_j^*}) = \int_0^\infty \exp\left(-\beta d^2_{i_j^*,\neg i_j^*;Y}\right) f\left(d^2_{i_j^*,\neg i_j^*;Y}\right) dd^2_{i^*,\neg i^*;Y} \quad (8)
$$

is the Laplace transform of lognormal distribution. Although, there is no closed form solution of (8), but[61] derives a sharp approximator of (8) for $\beta > 0$ using *Lambert's W function*. Therefore we propose the following Monte Carlo procedure to estimate $E(W_{i_j^*,\neg i_j^*})$ as follows:

a. Fit a standard geographically weighted regression with Gaussian Kernel in the response space and extract $\hat{\beta}$[62].
b. Fit the model (6) with $\log(d^2_{i_j^*,\neg i_j^*;Y})$ in the LHS and obtain $\hat{\mu}_{i_j^*,\neg i_j}$ and $\hat{\sigma}^2$.
c. Draw $R$ iid replicates of $d^2_{i_j^*,\neg i_j^*;Y}$ from lognormal$(0,\hat{\sigma}^2)$.
d. For each realization compute $\exp(-\hat{\beta}d^{2(r)}_{i_j^*,\neg i_j^*;Y}e^{\hat{\mu}_{i_j^*,\neg i_j^*}})$.
e. Then the Monte Carlo estimator of the LHS of (8) is given by
$\hat{E}(W_{i_j^*,\neg i_j^*}) = \frac{1}{R}\sum_{r=1}^R \exp(-\hat{\beta}d^{2(r)}_{i_j^*,\neg i_j^*;Y}e^{\hat{\mu}_{i_j^*,\neg i_j^*}})$While this Monte Carlo approximation works well when $\beta$ and $\sigma$ are small, it fails to explore the tail region as $\beta \to \infty$. Hence, if $\hat{\beta}$ is large, an efficient importance sampler, derived in[61], should be used.

### Ensemble topological regression

The above construction in (7) allows relaxing the disjointedness requirement $I^* \cap I = \phi$ to include the anchor points as neighborhood training points and allows modeling the $\tilde{W}'s$ through least squares optimization. However, by construction, $|I^*| < |I|$, meaning not all training points can be included as anchor points because the least squares model becomes overparameterized and overfits the training data leading to poor generalization performance. Since a subset of the available training set must be selected as anchor points, the results may be sensitive to the selected anchor points. To average out the effect of anchor points, one can simply randomly sample multiple different sets of anchor points and ensemble the results of each set. In order to achieve this, we introduce Ensemble TR, which samples $t$ sets of anchor points independently and generates average predictions from the resulting $t$ TR-models. The percentage of training instances to include as anchor instances can be viewed as a hyperparameter, so $t$ percentages can be sampled from a Gaussian distribution $\mathcal{N}(\mu_k,\sigma_k^2)$, with $\mu_k$ being the mean percentage of training instances to include as anchor instances and $\sigma_k^2$ being the requested variance of the $t$ percentages. To verify percentage values are valid and to prevent over or under-fitting, the sampled percentages are clipped between the range [30%, 90%]. This leaves the user with three parameters: the number of models ($t$), the mean percentage of training samples to include as anchor instances ($\mu_k$), and the variance of the percentages ($\sigma_k^2$). Ensemble TR maintains its computational efficiency considering $D_X^{N \times N}$ can be initially calculated, and $t D_X^{N \times K}$'s can be easily sampled from $D_X^{N \times N}$. This means that once distances are calculated, only $t$ multi-task linear regression models must be solved and RBF kernels applied to their outputs to generate predictions, leading to fast run times.

### Reporting summary

Further information on research design is available in the Nature Portfolio Reporting Summary linked to this article.

## Data availability

The ChEMBL datasets used in this study are available in the ChEMBL database (https://www.ebi.ac.uk/chembl/)[58]. The code to extract the 530 ChEMBL datasets is provided in the code repository. Source data are provided with this paper.

## Code availability

Sample data files and Python code to regenerate the TR figures and results are openly provided at https://github.com/Ribosome25/TopoReg_QSAR, which is archived in Zenodo under the identifier https://doi.org/10.5281/zenodo.10929477[63].

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

## Acknowledgements
This work was supported in part by the National Science Foundation under Grants Nos. 2007903 (received by RP) and 2007418 (Received by S.G) and Leidos Biomed/NCI under contract 22X049 Any opinions, findings, and conclusions or recommendations expressed in this mate-rial are those of the authors and do not necessarily reflect the views of the National Science Foundation or Leidos Biomed/NCI. The authors acknowledge the High Performance Computing Center (HPCC) at Texas Tech University for providing computational resources that have con-tributed to the research results reported within this paper. http://www.hpcc.ttu.edu.

## Author contributions
R.Z., D.N., S.G., and R.P. formulated the problem and conceived the experiments, R.Z., D.N., C.S., conducted the experiments, R.Z., D.N., C.S., S.G., and R.P. analyzed the results. All authors reviewed the manuscript. R.Z. conducted this work while he was working at Texas Tech University, however, he is currently working at Merck Inc.

## Competing interests
The authors declare no competing interests.
