## [Peer Review File · Nature Communications]

Topological Regression as an interpretable and efficient tool for Quantitative Structure-Activity Relationship ModelingREVIEWER COMMENTS

Reviewer #1 (Remarks to the Author):

The paper performs a large-scale study on the performance of different predictive modelling techniques on a set of about 530 public potency data sets from ChEMBL. The authors compare the performance of ChemProp (their reference benchmark) against two projection-based approaches: MLKR with KNN and a newly developed approach: Topological Regression (TR).

The topic is quite relevant and of high interest. The publication is professionally written, and I would like to see it published soon.

Having said that, I have some concerns about the study design (relevant benchmarks & comparators) and the explanation of some aspects / arguments:

- The authors demonstrate that their approach reaches a similar performance to ChemProp. They claim that their method has a significantly lower computational complexity. Given that Deep Learning techniques are well known to be computationally demanding, this is believable - it would nevertheless be beneficial if the authors would back their claims up with data (e.g., statistics on run times, memory requirements).

- Furthermore, if computational complexity is a critical concern, I am wondering why the authors did not chose a simpler shallow learning approach (e.g., random forests, SVMs as reference benchmark). I believe that the paper would benefit from adding such a comparison to simpler approaches.

- Given that the authors focus on the development of a projection-based approach, I am surprised that they do not compare their work against other, similar techniques (e.g., t-SNE, uMAP, MDS, Sammon Mapping, Generative Topographic Mapping, Self-Organizing Maps, Network-like Similarity Graphs, ...). A discussion of such methods is missing. I believe that the paper would benefit significantly from such a comparison as it would allow the reader to understand the advantages and disadvantages of the presented approach to more traditional techniques. If the authors consider such an analysis not relevant, they should at least mention and discuss alternative approaches and why they do not consider them relevant.

- Along these lines: the authors demonstrate the behavior of their method in a couple of examples. This can be instructive and support an intuitive understanding. At the same time, I am missing any systematic, quantitative study of the learned similarities / maps / spaces. As it is an important advantage of the method that compounds with comparable properties are grouped together better than by just considering their "chemical similarity", it would be beneficial to demonstrate quantitatively and systematically that this is indeed the case beyond a few selected examples. As the authors explicitly mention activity cliffs, it might be interesting to study how such activity cliffs behave in their method relative to other approaches in a systematic way. If their approach is indeed able to learn a mapping of chemical space that smoothed out a considerable number of activity cliffs, that would be of interest in my point of view. The SALI-related analyses in the supplementary material would deserve to become part of the main manuscript.

- Finally, it would be highly beneficial if the authors would explain in more detail what exactly they mean with "interpretability" (or: how they define it) in this context and how they envision their method to be used on lead identification and optimization approaches.

My understanding is that in LO campaigns, it is typically truly relevant for users to understand which parts of a compound are crucial for a given property / activity (or which aspects to change to optimize something). This is an actionable information that allows chemists to produce innovative ideas. The authors claim that "in the sense that, the set of "most important" features may not correspond to a chemically feasible molecule" - although such approaches are often used to highlight relevant substructures ("glowing molecules"). I would like to see a more balanced

discussion here.

This is particularly relevant as the authors later use a fingerprint-based method to calculate similarities between molecules. I may have missed it, but I would like to understand better what the conceptual difference between using a fingerprint to measure the similarity of compounds or to train a model directly is. The authors could clarify that.

Explainable AI is a highly relevant topic and of great interest to experts in the field. It may well be that projection-based approaches offer benefits over methods relying on feature importance or contributions. It should however be clear where those advantages (and drawbacks) are.

Reviewer #2 (Remarks to the Author):

This manuscript aims at probing a new proposed interpretable technique, so-called topological regression (TR), for the results of QSAR machine learning-based models. It compares the performance and efficiency of TR to other prediction approaches, such as Metric Learning Kernel Regression (MLKR) and ChemProp, and evaluates its usefulness in molecular design. The motivation for this study is well-explained, and the approach appears to be sound for the intended objectives. The conclusions drawn are supported by the obtained results. However, the manuscript requires a revision to address the following points:

1 Introduction:

. A wider literature overview concerning the subject should be given to show good orientation in wider consequences. Specifically, references to concepts like the Shapley additive explanation (SHAP), counterfactuals, and the Transformer-CNN approach should be included (e.g., Refs. [1-3]). More importantly, in so doing, the specific reasons for applying the proposed computational approach developed over other schemes (e.g.: regarding their effectiveness in big data, computational costs, and potential forward molecular design efforts) should emerge throughout the manuscript.

. The same applies regarding the problems caused by the presence of activity cliffs in QSAR modelling. That is, additional references on the subject, considering more recent works, should be included (see e.g.: Refs [4-6]).

2 Data description and problem motivation:

. The justification for choosing ChemProp, which has been primarily developed for molecular property predictions and not for drug discovery endeavours, as the baseline model instead of other models like Transformer-CNN (Ref. [3]) should be further sustained and supported by more than one reference.

3 Results:

. The results shown in Table 2, after relaxing the disjointedness requirement in TR, need further clarification. It is not entirely clear if the TR results are indeed better than MLKR regarding scaffold splitting of data, and the reasons for the reversal of signs related to the Spearman correlation values should be discussed.

. The evaluation of interpretability based on a comparison with a simple KNN scheme for a few particular cases may not be sufficient to demonstrate the superiority of the proposed technique over other prediction schemes. Additionally, Figures 5 and 6 show that both MLKR and TR provide similar results, though TR has the advantage of being less computational expensive.

. Even though offering information about the goodness-of-fit can be advantageous, since TR relies on linear multivariate statistical-based approaches, it is important to discuss the verification of parametric assumptions and the adequacy of the proposed TR modelling technique in relation to these assumptions.

. Finally, the authors should carefully revise the whole manuscript to correct typos and other minor shortcomings, such as correcting references [11], [12], and [21]. More importantly, to enhance the FAIRness (Findable, Accessible, Interoperable, and Reusable) of the proposed TR technique, it

is essential to provide the code on the platform GitHub (i.e., on [https://github.com/Ribosome25/TopoReg QSAR](https://github.com/Ribosome25/TopoReg_QSAR)).

To sum up, while the initial impression suggests that this study will provide an efficient scheme for interpreting and predicting ML non-linear models of big data to assist in molecular design, it primarily focuses on finding similarity outputs with its advantages and possible drawbacks. Furthermore, the unique importance of the topological regression approach based on ECFP4 descriptors for interpretability in QSAR modelling and related design endeavours is not explicitly demonstrated in this work, which is limited to a few particular case studies.

- [1] Rodríguez-Pérez, R. (2019) Interpretation of Compound Activity Predictions from Complex Machine Learning Models Using Local Approximations and Shapley Values; <https://doi.org/10.1021/acs.jmedchem.9b01101>
- [2] Karpov, P. (2020) Transformer-CNN: Swiss Knife for QSAR Modeling and Interpretation; <https://doi.org/10.1186/s13321-020-00423-w>
- [3] Wellawatte, G.P. et al. (2022) Model Agnostic Generation of Counterfactual Explanations for Molecules; <https://doi.org/10.1039/D1SC05259D>
- [3] Cruz-Monteaudo, M. et al. (2014) Activity Cliffs in Drug Discovery: Dr Jekyll or Mr Hyde?; <https://doi.org/10.1016/j.drudis.2014.02.003>
- [4] Stumpfe, D. (2019) Evolving Concept of Activity Cliffs; <https://doi.org/10.1021/acs.omega.9b02221>
- [5] Hu, H. et al. (2020) Simplified Activity Cliff Network Representations with High Interpretability and Immediate Access to SAR Information; <https://doi.org/10.1007/s10822-020-00319-9>

Reviewer #3 (Remarks to the Author):

The authors introduce a QSAR method termed 'topological regression' (TR). The method shows comparable performance to ChemProp and Metric Learning for Kernel Regression (MLKR). Like MLKR, TR is a kernel method, where activity is predicted based on estimated weights. This allows the identification of "neighbor instances" that have the highest influence on the prediction for specific test instances, which might be valuable for interpretability.

The paper is essentially a methods paper and as such, placing the method description at the end after the conclusion does not work well as notions and notations introduced there are an essential prerequisite for the result section. The authors explain the methods well and the novelty of their approach make it principally suitable for publication.

While the authors clearly explain and show how the method works and how identification of neighbor instances differs from straightforward KNN approaches, they do little in the way of actual interpretation of the results they show. Given, that the authors state this as a key advantage of their method, this part should be extended in a major revision.

Specific points:

I.052: "... trained on ... as the training set"

"as the training set" is redundant.

I.060: "RDKit descriptors are some popular algorithms"

Descriptors are not algorithms.

II.080-3: The authors need to make a better argument for their vision of interpretability. I can to some extent get the point they make later where they argue how 'intuitive (?) interpretation at instance level' for individual predictions might be beneficial but would not necessarily see that type of interpretability as superior or as intuitive for that matter. (As mentioned elsewhere, no practical example is given.)

II.119-31: There is some redundancy in the outline of the paper in the last paragraph of section 1 and the first paragraph of section 2

II.156: "Y_i is the numeric response variable..." for clarity, consider adding "...of the i-th training

sample"

II.170-2: Was a folded or unfolded version of the Morgan fingerprint used?

II.172-3: The authors mention the Tanimoto coefficient (TC) and the use of 1-TC as distance metric while discussing MLKR. While I see how this metric can be applied in TR, MLKR utilizes the Mahalanobis metric, a generalized form of the Euclidean distance. I do not understand how 1-TC is used for MLKR.

II.207-7: The authors emphasize two different strategies for splitting training/test data. However, results are only discussed for the random split, while arguably the scaffold split could be considered more relevant in practice, as this is more likely to indicate independence between training and test set.

I.242,I.263: Fig.2 is redundant as the same results for the two methods there are also shown in Fig.3. Furthermore, different colors are used for the methods in Fig.2 and 3.

II.247: In any meaningful organization of the manuscript, the method section should follow at this point. The result section cannot be understood without understanding the notions and notations introduced in the method section at least up to the introduction of the regression coefficient matrix B. At the very least, the reader should be referred to section 5.1 to enable an understanding of the results section.

I.277,I.298: These tables also present the relative performance differences of the scaffold split. I would also suggest that the authors present the individual performances of the compared methods for the scaffolds split in a figure analogous to figure 3.

II.310-400: The interpretation section demonstrates the method only on a single example, showing actual structures of nearest neighbors only in figure 4. However, except for stating that the neighbors identified by TR belong to different scaffolds, no further interpretation is carried out. As interpretability is advertised as one of the main advantages of this method, the authors should extend the discussion based on the structures and demonstrate the benefit of their approach in practical terms. I would also expect a similar treatment for the results shown in Fig. 5 and 6.

I.491f: It is slightly irritating that on the one hand N^* is used to denote the number of all training samples while I^* only refers to the indices of the training set. A different superscript might have been used in the case of N^*

Reviewer #4 (Remarks to the Author):

Zhang et al. report a method for QSAR modeling that utilizes similarity-based regression. The distance in response space is modeled using chemical space distances. The authors evaluate this method on activity prediction for 530 ChEMBL targets. Even though the results are clearly presented, and the method might be of interest, it is difficult to assess its novelty. Related work in the field is almost not cited, and the work is not put in context of previous research. Another general concern is that the benefit of this technique compared to the state of the art is not clear. Very small differences in performance are observed compared to a black-box method for >500 targets. However, baselines of more standard algorithms that are also interpretable are not included. Authors mentioned the potential benefit of better predicting activity cliffs, but this is not clearly shown in the results. Other comments are listed below.

- Model performance. It should be clearer in the main text how the authors went from Fig.3 (where TR provides worse results) to Table 2 (where superior performance is obtained for TR). Also, comparisons per individual targets would be informative. For how many ChEMBL targets is TR better than the baseline models?

- Authors indicate that ChemProp does not rely on pre-defined features. This statement is incorrect. One needs to define atom/bond features.

- 'Standard shallow learners (RF, SVM) can offer feature importance scores, but such scores may not be chemically interpretable' – This statement is also incorrect. The importance scores can be chemically interpretable depending on the molecular representation utilized, this does not depend on the algorithm selection itself (if it can rely on different features). There are many RF/SVM-based QSAR models that are interpretable or explainable. Authors should mention this and cite previous work in the field.
- More information about molecules and activity data and its preparation should be provided, e.g. authors only refer to 'pChEMBL values' and indicate that 'SMILES string could not be converted to RDKit molecules were also removed'. Why they could not be converted? Where there any filtering or standardization steps applied for activity and molecular data?
- More information on the ChemProp model should be provided. For instance, instead of 'We use ChemProp's own hyperparameter search tool', authors could mention which hyperparameters were optimized and how.
- 'Anchor points' is mentioned in Section 3.1 for the first time, and it has not been defined.
- How does this work compare to generative topographic mapping (GTM) (applied to QSAR by A. Varnek et al.)?

Detailed Response to Reviewer Comments: Topological Regression in Quantitative Structure-Activity Relationship Modeling

The authors would like to thank all 4 Reviewers for their constructive review of the manuscript and their valuable comments.

Regarding the productive concerns raised by **Reviewer 1**, the following changes have been incorporated in the revised manuscript:

The paper performs a large-scale study on the performance of different predictive modelling techniques on a set of about 530 public potency data sets from ChEMBL. The authors compare the performance of ChemProp (their reference benchmark) against two projection-based approaches: MLKR with KNN and a newly developed approach: Topological Regression (TR). The topic is quite relevant and of high interest. The publication is professionally written, and I would like to see it published soon. Having said that, I have some concerns about the study design (relevant benchmarks & comparators) and the explanation of some aspects / arguments:

- The authors demonstrate that their approach reaches a similar performance to ChemProp. They claim that their method has a significantly lower computational complexity. Given that Deep Learning techniques are well known to be computationally demanding, this is believable - it would nevertheless be beneficial if the authors would back their claims up with data (e.g., statistics on run times, memory requirements).

As suggested by the reviewer, we have incorporated a subsection 4.2 in the results showing the computation times and memory requirements of the competing methods, with TR being much more efficient and less demanding. The summary table (Table 2 in revised manuscript) is shown below.

Table 2: Computational Comparison of competing methods showing train times, test times, and peak RAM consumption on the scaffold split.

Method	CPU System (AMD EPYC 7702, 2.0 GHz, 64 Physical Cores, 128 Logical Cores, 512 GB RAM)			GPU System (Intel Xeon Gold 6242, 2.8 GHz, 16 Physical Cores, 32 Logical Cores, 384 GB RAM, NVIDIA Tesla V100)		
	MLKR	TR	Ensemble TR	ChemProp	TCNN	TCNN Aug
Train Time (s)	181.65	1.602	13.768	53.831	30.586	109.759
Test Time (s)	0.354	1.008	12.301	0.509	6.915	20.003
Peak RAM (GB)	1.7768	0.253	0.314	2.044	2.864	3.231

- Furthermore, if computational complexity is a critical concern, I am wondering why the authors did not chose a simpler shallow learning approach (e.g., random forests, SVMs as reference benchmark). I believe that the paper would benefit from adding such a comparison to simpler approaches.

We have updated the manuscript to include random forest as a baseline model in the results table, which shows TR and TCNN are able to achieve better performance than RF. The summary table (Table 1 in revised manuscript) is shown below.

Table 1: NRMSE and Spearman's ρ of competing methods on both random and scaffold split

	CV Split		Scaffold	
	Spearman	NRMSE	Spearman	NRMSE
RF	0.7629	0.6242	0.6493	0.7395
MLKR	0.7487	0.6420	0.6342	0.7619
ChemProp	0.7160	0.6776	0.5986	0.8002
TCNN	0.7437	0.6595	0.6321	0.7692
TCNN Aug	0.7858	0.5961	0.6742	0.7176
TR*	0.6935	0.7023	0.5675	0.8255
TR	0.7625	0.6255	0.6574	0.7330
Ensemble TR	0.7847	0.5989	0.6791	0.7101

- Given that the authors focus on the development of a projection-based approach, I am surprised that they do not compare their work against other, similar techniques (e.g., t-SNE, uMAP, MDS, Sammon Mapping, Generative Topographic Mapping, Self-Organizing Maps, Network-like Similarity Graphs, ...). A discussion of such methods is missing. I believe that the paper would benefit significantly from such a comparison as it would allow the reader to understand the advantages and disadvantages of the presented approach to more traditional techniques. If the authors consider such an analysis not relevant, they should at least mention and discuss alternative approaches and why they do not consider them relevant.

As pointed out by the reviewer, we have added a discussion about other projection-based techniques in the introduction and addressed how these methods are unsupervised besides MLKR, which helps it smoothen activity cliffs. To further this point, we included a figure showing how the unsupervised projection methods lead to rough response surfaces, whereas MLKR was able to obtain a smoother response space (Fig 1 in revised manuscript and shown below). We also added more clarity to point out that TR is inspired by the projection technique MLKR, but isn't a method for dimensionality reduction, rather it predicts the distances of anchor points to the query sample in the response space. Additionally, we extended the interpretability of TR through KNN-graphs, which are very similar to Network-like Similarity Graphs (NSGs). Therefore, we also included NSGs and other similarity-based network methods in the introduction. NSGs are a method to visualize the similarity of molecules, lead optimization pathways, and activity cliffs for target datasets, whereas TR is a method to perform similarity-based regression. We show however, that we can visualize TR predictions in the form of KNN-graphs that are similar to NSGs.

Fig. 1: The transformed chemical space of target CHEMBL4530 using various projection methods. Notice how activity cliffs are present regardless of the projection method. Additionally, notice that MLKR creates the best separation of the two similar (Tanimoto Similarity = 0.69) molecules, CHEMBL2086505 (pChEMBL=7.41) and CHEMBL2086502 (pChEMBL=4.8).

- Along these lines: the authors demonstrate the behavior of their method in a couple of examples. This can be instructive and support an intuitive understanding. At the same time, I am missing any systematic, quantitative study of the learned similarities / maps / spaces. As it is an important advantage of the method that compounds with comparable properties are grouped together better than by just considering their "chemical similarity", it would be beneficial to demonstrate quantitatively and systematically that this is indeed the case beyond a few selected examples. As the authors explicitly mention activity cliffs, it might be interesting to study how such activity cliffs behave in their method relative to other approaches in a systematic way. If their approach is indeed able to learn a mapping of chemical space that smoothed out a considerable number of activity cliffs, that would be of interest in my point of view. The SALI-related analyses in the supplementary material would deserve to become part of the main manuscript.

We thank the review for pointing this out. We furthered the interpretability analysis on a dataset level by forming KNN-graphs (as shown in Fig 6 of revised manuscript and shown below) based on the predictions of the different similarity methods. This allowed clustering the samples based on their predictions and viewing how homogenous the clusters and predictions were for a whole dataset. We then calculated the average within-cluster standard deviations for all 530 datasets for a quantitative measure of activity smoothness (Fig 7 of revised manuscript). These were plotted as line graphs allowing a glimpse at how TR behaves compared to other similarity-based methods across all datasets. Lastly, we extended the interpretability discussion through visualization of the clusters formed from TR KNN-graphs. We show how, synonymous with NSG, the clusters can be traversed and used in lead optimization, lead identification, and

for general design ideas (Section 4.3 Interpreting TR in revised manuscript). Given the extension of the interpretability discussion and figure/table limit, we have kept the SALI analysis in the supplementary material.

Fig. 6: KNN-Graphs visualizing the 5 nearest neighbor test predictions of KNN, MLKR, and TR for a single CV fold of the foregoing target activity Phospholipase D2 (CHEMBL2734).

Fig. 7: Quantitative analysis across all 530 datasets showing the average within-cluster standard deviation obtained from the test prediction KNN-graphs for the competing methods.

- Finally, it would be highly beneficial if the authors would explain in more detail what exactly they mean with "interpretability" (or: how they define it) in this context and how they envision their method to be used on lead identification and optimization approaches.

We thank the review for this comment, we have strengthened the introduction by providing a definition of model interpretability and common methods used in QSAR modeling to achieve it. Additionally, as mentioned above, we also furthered the interpretability section through KNN-graphs, similar to NSG, which, as we show, can be used for lead identification, lead optimization pathway visualization, and general design ideas (section 4.3. in revised manuscript and optimization pathway example of revised Figure 8 included next). We envision a chemist can utilize the training clusters to find lead chemicals with other desired properties, visualize previous optimization pathways to see what changes resulted in the specific outcomes, and easily find innovative design ideas, such as analyzing common molecular structures in the active or inactive clusters.

Fig. 8: Optimization pathway visualization in the most active training cluster of target protein complex Integrin alpha-4/beta-7 (ChEMBL278). a) depicts the training neighborhood graph obtained from TR predictions, b) depicts the minimum spanning tree of the most active cluster with a minimum path connecting the most active and least active molecules, c) 5 example molecules showing the lead optimization pathway.

My understanding is that in LO campaigns, it is typically truly relevant for users to understand which parts of a compound are crucial for a given property / activity (or which aspects to change to optimize something). This is an actionable information that allows chemists to produce innovative ideas. The authors claim that "in the sense that, the set of "most important" features may not correspond to a chemically feasible molecule" - although such approaches are often used to highlight relevant substructures ("glowing molecules"). I would like to see a more balanced discussion here.

As suggested by the reviewer, we have altered this paragraph to provide a more balanced discussion. When working with interpretable descriptors, knowing the set of most important features can provide relevant substructures but they are inadequate for mapping directly to feasible molecules. Additionally, they don't answer the question "What changes will result in an alternate outcome?". Furthering this point, we have added a discussion on SHAP and MMACE which have their own advantages and drawbacks acknowledged, such as being model agnostic, but also being reliant on strong model generalization as the methods are based on the model's knowledge.

This is particularly relevant as the authors later use a fingerprint-based method to calculate similarities between molecules. I may have missed it, but I would like to understand better what the conceptual difference between using a fingerprint to measure the similarity of compounds or to train a model directly is. The authors could clarify that.

TR models the distance between samples in the response space using the distance in the predictor space. Conceptually the difference is that a model trained on fingerprints, x , is a function of those fingerprints, $f(x)$, whereas training a model on fingerprints distances, $d(x_1, x_2)$, results in a function of those distances, i.e. $f(d(x_1, x_2))$. The distance doesn't have to be calculated from a fingerprint but can be empirically

designed or extracted from any representation, lending more leeway to TR as all it needs are pre-defined distances/similarities and not direct fingerprints. Additionally, using the similarity compared to the full fingerprint allows reducing the complexity of the modeling method, which would be beneficial for small samples sizes, turning a problem where the number of samples is less than the number of model parameters into a problem where the number of samples is greater than the predictor space as $|I^| < |I|$.*

Explainable AI is a highly relevant topic and of great interest to experts in the field. It may well be that projection-based approaches offer benefits over methods relying on feature importance or contributions. It should however be clear where those advantages (and drawbacks) are.

We have included more explanation to the advantage of similarity-based methods compared to other interpretability methods, specifically as similarity-based methods directly provide samples which are influencing the model's prediction the most. An alternative to QSAR is read-across, which is entirely similarity based and used by many chemists as it allows quickly comparing analogous neighbors and analyzing the validity through their expert knowledge. Similarity based methods also allow similarity-based graphs, like NSGs, to be constructed and analyzed which can be valuable for analyzing activity cliffs and pathways. We have extended the interpretability section to show how TR can be used for clustering through KNN-graphs, which highlights how it smooths out activity cliffs and how it can be used in lead identification/optimization. Also, TR can be paired with Laplacian Scores, MMACE or SHAP adding additional layers to its interpretability by allowing global feature importance, prediction-wise feature importance, and unseen counterfactuals.

*Regarding the productive concerns raised by **Reviewer 2**, the following changes have been incorporated in the revised manuscript:*

This manuscript aims at probing a new proposed interpretable technique, so-called topological regression (TR), for the results of QSAR machine learning-based models. It compares the performance and efficiency of TR to other prediction approaches, such as Metric Learning Kernel Regression (MLKR) and ChemProp, and evaluates its usefulness in molecular design. The motivation for this study is well-explained, and the approach appears to be sound for the intended objectives. The conclusions drawn are supported by the obtained results. However, the manuscript requires a revision to address the following points:

1 Introduction: A wider literature overview concerning the subject should be given to show good orientation in wider consequences. Specifically, references to concepts like the Shapley additive explanation (SHAP), counterfactuals, and the Transformer-CNN approach should be included (e.g., Refs. [1-3]). More importantly, in so doing, the specific reasons for applying the proposed computational

approach developed over other schemes (e.g.: regarding their effectiveness in big data, computational costs, and potential forward molecular design efforts) should emerge throughout the manuscript.

. The same applies regarding the problems caused by the presence of activity cliffs in QSAR modelling. That is, additional references on the subject, considering more recent works, should be included (see e.g.: Refs [4-6]).

As suggested by the Reviewer, a wider literature review was conducted, and the suggested references were addressed in the paper. We included additional references as well such as Network-like Similarity Graphs, and other similarity-based methods to further our reasoning and provide more background. We additionally added Transformer-CNN as a stronger baseline model and showed how TR achieves better results on scaffold split and comparable performance on the random split.

2 Data description and problem motivation:

. The justification for choosing ChemProp, which has been primarily developed for molecular property predictions and not for drug discovery endeavours, as the baseline model instead of other models like Transformer-CNN (Ref. [3]) should be further sustained and supported by more than one reference.

In addition to adding Transformer-CNN as a baseline model as suggested, we have added two additional references which highlight ChemProp's use and success in QSAR modeling. Specifically, 'Deep learning-guided discovery of an antibiotic targeting Acinetobacter baumannii' (doi: 10.1038/s41589-023-01349-8) utilized ChemProp for antibiotic discovery targeting acinetobacter baumannii, and 'Machine Learning for Fast, Quantum Mechanics-Based Approximation of Drug Lipophilicity' (doi: 10.1021/acsomega.2c05607) which shows ChemProp to perform the best out of a variety of models(Random Forest, Lasso, XGBoost, Chemprop, and Chemprop3D) at predicting drug lipophilicity, which is important in early-stage drug discovery.

3 Results: The results shown in Table 2, after relaxing the disjointedness requirement in TR, need further clarification. It is not entirely clear if the TR results are indeed better than MLKR regarding scaffold splitting of data, and the reasons for the reversal of signs related to the Spearman correlation values should be discussed.

We agree the presentation of the results could lead to confusion. We decided to change the results section to not include relative performances and include only one table showing all competing methods true performances. We detailed the names and difference between each TR method, i.e. before relaxing the disjointedness requirement, TR, and after relaxing the disjointedness requirement, TR. We also improved upon the performance of TR by including an ensemble of TR models with different anchor sets, and introduced this method as Ensemble TR. We also changed the percentage of anchor points from 20% to 50% for TR, which improved its results relative to MLKR and other methods.*

Table 1: NRMSE and Spearman’s ρ of competing methods on both random and scaffold split

	CV Split		Scaffold	
	Spearman	NRMSE	Spearman	NRMSE
RF	0.7629	0.6242	0.6493	0.7395
MLKR	0.7487	0.6420	0.6342	0.7619
ChemProp	0.7160	0.6776	0.5986	0.8002
TCNN	0.7437	0.6595	0.6321	0.7692
TCNN Aug	0.7858	0.5961	0.6742	0.7176
TR*	0.6935	0.7023	0.5675	0.8255
TR	0.7625	0.6255	0.6574	0.7330
Ensemble TR	0.7847	0.5989	0.6791	0.7101

The evaluation of interpretability based on a comparison with a simple KNN scheme for a few particular cases may not be sufficient to demonstrate the superiority of the proposed technique over other prediction schemes. Additionally, Figures 5 and 6 show that both MLKR and TR provide similar results, though TR has the advantage of being less computational expensive.

We thank the reviewer for pointing this out and agree that specific test cases aren’t enough to show the superiority. Therefore, as mentioned above, we furthered this analysis using KNN-graphs to show, on a dataset level, how TR predictions lead to more homogenous clusters of activity compared to other similarity-based methods. Additionally, to demonstrate this behavior for all datasets, we calculated the average within-cluster standard deviations for each method. The results for each dataset were plotted on a line graph to show how TR achieved more homogenous clusters across all datasets (Fig 7 of revised manuscript). Also, we furthered this analysis by visualizing the training clusters and expanded on how these homogenous clusters could be utilized for lead optimization/identification and for mining innovative design ideas. In addition, TR can be paired with Laplacian Scores, SHAP or MMACE. Training sample importance offers another view often not available for non-similarity-based methods, which is beneficial to practitioners, just as it has been in read-across for decades. Lastly, since we had to add a few figures to show the above points, we relegated figures 5 and 6 to the supplementary.

Even though offering information about the goodness-of-fit can be advantageous, since TR relies on linear multivariate statistical-based approaches, it is important to discuss the verification of parametric assumptions and the adequacy of the proposed TR modelling technique in relation to these assumptions.

This is a good question. Since the standard coefficient of determination offers immediate goodness-of-fit statistic for linear models (or transformed linear models), we compute the training R -sq values for all the 530 ChemBL datasets considered here. However, we point out that the linear model connects W_{ij} ‘s with d^2_{ij} ‘s and hence all the reported adequacy measures correspond to the system of equations outlined in Eq(3) in the main manuscript. The average R -sq turns out to be 0.8396. Evidently, our conceptually straightforward parametric linear model has sufficient power to explain variation in W_{ij} .

Since R -sq is calculated on training data only, it does not offer predictive adequacy. To assess that, we compute the prediction interval for each test point in the cross-validation set. Once again, the linear model specification allows us to compute the prediction interval analytically. We then compute the coverage of these prediction intervals across all the folds and compute the coverage. Ideally, we would like to see the coverage of the prediction interval achieves nominal level. In all the 530 datasets across all the folds, the

coverage of 95% prediction interval is 94.31%. Clearly, the model specified in Eq (3) is adequate for prediction purpose as well. Below we offer plots of the observed Vs predicted distances in the response space with the prediction interval overlaid for a few randomly chosen ChemBL dataset.

Figure S5: Scatter plot of predicted vs observed response distances of target CHEMBL4530 with 95% prediction interval.

Figure S6: Scatter plot of predicted vs observed response distances of target CHEMBL2734 with 95% prediction interval

. Finally, the authors should carefully revise the whole manuscript to correct typos and other minor shortcomings, such as correcting references [11], [12], and [21]. More importantly, to enhance the FAIRness (Findable, Accessible, Interoperable, and Reusable) of the proposed TR technique, it is essential to provide the code on the platform GitHub

(i.e., on <https://github.com/Ribosome25/TopoReg> QSAR).

We thank the Reviewer for pointing those out, we have proofread the manuscript for typos and corrected the mentioned references. In addition, we have updated the GitHub page to include all code to generate the main results and figures as well as sample ChEMBL datasets.

To sum up, while the initial impression suggests that this study will provide an efficient scheme for interpreting and predicting ML non-linear models of big data to assist in molecular design, it primarily focuses on finding similarity outputs with its advantages and possible drawbacks. Furthermore, the unique importance of the topological regression approach based on ECFP4 descriptors for interpretability in QSAR modelling and related design endeavours is not explicitly demonstrated in this work, which is limited to a few particular case studies.

- [1] Rodríguez-Pérez, R. (2019) Interpretation of Compound Activity Predictions from Complex Machine Learning Models Using Local Approximations and Shapley Values; <https://doi.org/10.1021/acs.jmedchem.9b01101>
- [2] Karpov, P. (2020) Transformer-CNN: Swiss Knife for QSAR Modeling and Interpretation; <https://doi.org/10.1186/s13321-020-00423-w>
- [3] Wellawatte, G.P. et al. (2022) Model Agnostic Generation of Counterfactual Explanations for Molecules; <https://doi.org/10.1039/D1SC05259D>
- [3] Cruz-Monteagudo, M. et al. (2014) Activity Cliffs in Drug Discovery: Dr Jekyll or Mr Hyde?; <https://doi.org/10.1016/j.drudis.2014.02.003>
- [4] Stumpfe, D. (2019) Evolving Concept of Activity Cliffs; <https://doi.org/10.1021/acsomega.9b02221>
- [5] Hu, H. et al. (2020) Simplified Activity Cliff Network Representations with High Interpretability and Immediate Access to SAR Information; <https://doi.org/10.1007/s10822-020-00319-9>

Regarding the productive concerns raised by Reviewer 3, the following changes have been incorporated in the revised manuscript:

The authors introduce a QSAR method termed 'topological regression' (TR). The method shows comparable performance to ChemProp and Metric Learning for Kernel Regression (MLKR). Like MLKR, TR is a kernel method, where activity is predicted based on estimated weights. This allows the identification of "neighbor instances" that have the highest influence on the prediction for specific test instances, which might be valuable for interpretability. The paper is essentially a methods paper and as such, placing the method description at the end after the conclusion does not work well as notions and notations introduced there are an essential prerequisite for the result section. The authors explain the methods well and the novelty of their approach make it principally suitable for publication. While the authors clearly explain and show how the method works and how identification of neighbor instances differs from straightforward KNN approaches, they do little in the way of actual interpretation of the results they show. Given, that the authors state this as a key advantage of their method, this part should be extended in a major revision.

We thank the Reviewer for these constructive comments. As suggested, we have moved the methods section to appear before the results section for proper flow of the paper. We have improved upon the initial TR results by incorporating an ensemble of TR models with different anchor sets. We have also extended

the interpretability analysis using KNN-graphs to show how TR predictions compare to the other methods on a whole dataset level, rather than for a few test cases. We then used these graphs and their clusters to show how TR achieves more homogenous clusters of activity across all datasets by displaying the average within-cluster standard deviation for the competing methods. Finally, to further the interpretability, we showed how these clusters could be used for lead identification and optimization as well as pathway visualization.

Specific points:

I.052: "... trained on ... as the training set" "as the training set" is redundant.

I.060: "RDKit descriptors are some popular algorithms" Descriptors are not algorithms.

We thank the Reviewer for bringing these to our attention, these sentences have been updated to fix these errors.

II.080-3: The authors need to make a better argument for their vision of interpretability. I can to some extent get the point they make later where they argue how 'intuitive (?) interpretation at instance level' for individual predictions might be beneficial but would not necessarily see that type of interpretability as superior or as intuitive for that matter. (As mentioned elsewhere, no practical example is given.)

In the revised manuscript, we have extended the discussion on the advantages of similarity-based methods. We have included a small discussion on read-across, which are similarity-based methods that are often used as an alternative to QSAR models for property prediction due to their interpretability. In addition to naturally providing training instance importance, similarity-based methods allow informative visualizations of the drug-target landscape through Network-like similarity graphs (NSGs), and off-target interactions through methods like SAE and CSNAP. Using KNN-graphs, which visualize the predictions of the similarity-based methods similar to NSG, we extended the TR interpretability section to show how TR KNN-graphs can be used in lead identification, lead optimization, and for mining design ideas. Additionally, TR can be paired with SHAP or MMACE to provide feature importance or to generate unseen counterfactuals. Also, through the KNN-graphs, Laplacian scores can be calculated for global feature importance. On top of the training instance importance, which is often difficult for non-similarity-based models, this allows multiple layers of interpretability for practitioners to utilize.

II.119-31: There is some redundancy in the outline of the paper in the last paragraph of section 1 and the first paragraph of section 2

We agree that the paragraphs are redundant, we have removed the last paragraph from section 1 to remove the redundancy and save space.

II.156: "Y_i is the numeric response variable..." for clarity, consider adding "...of the i-th training sample"
As suggested, we have added "of the i-th training sample" to improve the clarity of the sentence.

II.170-2: Was a folded or unfolded version of the Morgan fingerprint used?

Section 2 has been updated with more details regarding the data extraction protocol where we have added that the folded ECFP4 descriptors were calculated and included the bit-size used (1024).

II.172-3: The authors mention the Tanimoto coefficient (TC) and the use of 1-TC as distance metric while discussing MLKR. While I see how this metric can be applied in TR, MLKR utilizes the Mahalanobis metric, a generalized form of the Euclidean distance. I do not understand how 1-TC is used for MLKR. The authors now see how this would lead to confusion, the use of TC and 1-TC is meant for TR only and not MLKR. We moved the ECFP4 and TC introduction out of the MLKR paragraph and included it with the data extraction paragraphs.

II.207-7: The authors emphasize two different strategies for splitting training/test data. However, results are only discussed for the random split, while arguably the scaffold split could be considered more relevant in practice, as this is more likely to indicate independence between training and test set. As suggested by the reviewer, we have included the scaffold split results as boxplots and discuss both random and scaffold split results in this paragraph.

I.242,I.263: Fig.2 is redundant as the same results for the two methods there are also shown in Fig.3. Furthermore, different colors are used for the methods in Fig.2 and 3. We thank the Reviewer for pointing this out, we have removed figure 2 and now reference the updated figure 3(4 in revised manuscript) which contains all competing methods as well as scaffold split.

II.247: In any meaningful organization of the manuscript, the method section should follow at this point. The result section cannot be understood without understanding the notions and notations introduced in the method section at least up to the introduction of the regression coefficient matrix B. At the very least, the reader should be referred to section 5.1 to enable an understanding of the results section. As suggested by multiple Reviewers, the Methods section has been moved to appear before the results section for improving the flow of the paper.

I.277,I.298: These tables also present the relative performance differences of the scaffold split. I would also suggest that the authors present the individual performances of the compared methods for the scaffolds split in a figure analogous to figure 3. As mentioned earlier, we have included the scaffold split boxplots in addition to the random split boxplots in Figure 4.

II.310-400: The interpretation section demonstrates the method only on a single example, showing actual structures of nearest neighbors only in figure 4. However, except for stating that the neighbors identified by TR belong to different scaffolds, no further interpretation is carried out. As interpretability is advertised as one of the main advantages of this method, the authors should extend the discussion based on the structures and demonstrate the benefit of their approach in practical terms. I would also expect a similar treatment for the results shown in Fig. 5 and 6.

We thank the reviewer for this comment and agree that an extension on the interpretability of predictions was needed. As mentioned earlier, we extended the interpretability discussion using KNN-graphs to show, for a whole dataset, how TR makes smoother predictions compared to other similarity-based approaches.

Additionally, we use the within-cluster deviations to show, for all datasets, that TR produces more homogeneous clusters of activity. Lastly, to extend the discussion further, we showed how the KNN-graphs allow clustering of the training dataset and test predictions, which can be mined by practitioners for design ideas, lead molecules, and can be used to visualize optimization pathways in the same way as NSGs. With the added interpretability figures and figure limit, we have moved Figures 5 and 6 to the supplementary materials.

I.491f: It is slightly irritating that on the one hand N^* is used to denote the number of all training samples while I^* only refers to the indices of the training set. A different superscript might have been used in the case of N^*

The authors thank the reviewer for pointing this out, we have adjusted N^ to properly maintain I^* 's relation as the subset associated to the anchor points.*

*Regarding the productive concerns raised by **Reviewer 4**, the following changes have been incorporated in the revised manuscript:*

Zhang et al. report a method for QSAR modeling that utilizes similarity-based regression. The distance in response space is modeled using chemical space distances. The authors evaluate this method on activity prediction for 530 ChEMBL targets. Even though the results are clearly presented, and the method might be of interest, it is difficult to assess its novelty. Related work in the field is almost not cited, and the work is not put in context of previous research. Another general concern is that the benefit of this technique compared to the state of the art is not clear. Very small differences in performance are observed compared to a black-box method for >500 targets. However, baselines of more standard algorithms that are also interpretable are not included. Authors mentioned the potential benefit of better predicting activity cliffs, but this is not clearly shown in the results. Other comments are listed below.

We thank the Reviewer for these constructive comments. We have conducted a wider literature review and included more related work citations. Specifically, we have included SHAP and MMACE for QSAR interpretability, and Transformer-CNN as a stronger baseline model. In addition, we have incorporated more similarity based QSAR methods such as network-like Similarity Graphs (NSGs), Similarity-Ensemble Approach (SEA), and Chemical Similarity Network Analysis Pull-down (CSNAP) for their interpretability through visualizing the drug-target interactions and activity cliffs. We have also achieved greater predictive performance by introducing ensemble TR, which ensembles the predictions of multiple TR models with different anchor sets. To further the point that TR smooths out of activity cliffs and to further the interpretability, we displayed the clusters formed by TR predictions using KNN-graphs. We then used the KNN-graphs to depict the interpretability on a dataset level, rather than just for particular test cases. Using the clusters formed by the graphs, we calculated the average within-cluster standard deviation for each dataset. These results were plotted as line graphs to depict that TR achieves more homogenous

clusters of activity, therefore leading to smoother activity cliffs. Lastly, to further the interpretability, we showed how the TR predictions with KNN-Graphs could be used by chemists for lead identification/optimization through visualizing pathways and mining the clusters for relevant patterns.

- **Model performance.** It should be clearer in the main text how the authors went from Fig.3 (where TR provides worse results) to Table 2 (where superior performance is obtained for TR). Also, comparisons per individual targets would be informative. For how many ChEMBL targets is TR better than the baseline models?

The results changed from table 1 to table 2 as we relaxed the disjointness requirement. The authors agree that the 2 separate relative performance tables in the results section could lead to confusion. We have modified the results section to include only one table with true performance of each method. We specified TR* as the method with disjoint anchor and neighborhood training sets, and TR as the relaxed disjointness requirement method. In addition, we devised a method, Ensemble TR, to ensemble multiple TR models with different anchor sets, which lead to improved generalization performance over other competing methods. All of this is detailed in the revised results section. In the supplementary material, we have provided a figure to depict the competing model performance vs. sample size. To do so we quantized the 530 datasets into 10 quantiles based on sample size and took the average NRMSE for each method for each bin (as shown in supplementary figure S7). In this section we also provide that TR Ensemble outperforms TCNN with augmentation on 293 of the 530 total target datasets and 424 of the 530 datasets when augmentation is not performed.

Figure S7: Visualizing the effect of sample size for the competing methods. The target datasets were binned into 10 quantiles based on sample size and the average NRMSE for each competing method was calculated and displayed for the scaffold split.

- **Authors indicate that ChemProp does not rely on pre-defined features. This statement is incorrect. One needs to define atom/bond features.**

We have updated the sentence to reflect this.

- ‘Standard shallow learners (RF, SVM) can offer feature importance scores, but such scores may not be chemically interpretable’ – This statement is also incorrect. The importance scores can be chemically interpretable depending on the molecular representation utilized, this does not depend on the algorithm selection itself (if it can rely on different features). There are many RF/SVM-based QSAR models that are interpretable or explainable. Authors should mention this and cite previous work in the field.

The authors would like to thank the reviewer for pointing this out. We have revised this entire paragraph to fix multiple errors and include numerous citations regarding the interpretability. One utilizing RFs and SVMs for QSAR interpretability and others including SHAP and MMACE as alternative QSAR interpretability methods. Additionally, we have provided a definition of interpretability for clarity and expanded on multiple similarity-based interpretability methods and their advantages, especially for visualization.

- More information about molecules and activity data and its preparation should be provided, e.g. authors only refer to ‘pChEMBL values’ and indicate that ‘SMILES string could not be converted to RDKit molecules were also removed’. Why they could not be converted? Where there any filtering or standardization steps applied for activity and molecular data?

We have included more details pertaining to the ChEMBL data extraction protocol and ECFP4 extraction which utilized RDKit and requires no standardization steps. Also, some SMILES strings were chemically invalid (the same as described in this article <https://www.ncbi.nlm.nih.gov/pmc/articles/PMC9926805/>), meaning that when given to RDKit, they result in errors. We have included that the ‘chemically invalid SMILES strings’ which could not be converted were removed.

- More information on the ChemProp model should be provided. For instance, instead of ‘We use ChemProp’s own hyperparameter search tool’, authors could mention which hyperparameters were optimized and how.

We have updated the sentence to include which hyperparameters were optimized and mention that Bayesian optimization was employed.

- ‘Anchor points’ is mentioned in Section 3.1 for the first time, and it has not been defined.

As suggested by multiple reviewers, the authors have moved the methods section to appear before the results as we agree this impacts the flow of the manuscript.

- How does this work compare to generative topographic mapping (GTM) (applied to QSAR by A. Varnek et al.)?

In comparison to GTM, TR is a similarity-based regression framework, while GTM is an unsupervised projection-based approach, often used for dimensionality reduction. TR does not attempt to provide a lower-dimensional projection, rather it provides predicted distances between anchor samples and the query sample in the response space. We have incorporated a few sentences comparing MLKR and TR to other unsupervised projection-based approaches and included a figure (Fig 1 of revised manuscript) in the

introduction to show how MLKR, which is supervised, leads to smoother activity cliffs compared to other unsupervised projection approaches like GTM, MDS, UMAP, t-SNE, which helped motivate the TR formulation.

Fig. 1: The transformed chemical space of target CHEMBL4350 using various projection methods. Notice how activity cliffs are present regardless of the projection method. Additionally, notice that MLKR creates the best separation of the two similar (Tanimoto Similarity = 0.69) molecules, CHEMBL2086505 (pChEMBL=7.41) and CHEMBL2086502 (pChEMBL=4.8).

REVIEWER COMMENTS

Reviewer #1 (Remarks to the Author):

I have read the author's response to the reviewer's comments and it seems the most important topics are addressed. I don't have any urgent additional comments.

Reviewer #2 (Remarks to the Author):

The authors have effectively addressed all the comments from the reviewers and carefully revised the manuscript, alongside with providing now the code accessible in the platform GitHub. Therefore, the revised manuscript can be published in its current form.

Reviewer #2 (Remarks on code availability):

The code provides an appropriate README file with a detailed introduction, and the extent the results of the paper are reproducible is guaranteed. The code, along with its scripts, is a usable resource for the QSAR community.

Reviewer #3 (Remarks to the Author):

The authors present a much improved version of their manuscript addressing all points raised by the reviewers.

I consider the revised publication acceptable for publication.

A few minor issues should be fixed:

Figure 1: Shows a smooth interpolated activity surface based on the activities of the projected molecules. For clarity, the authors could clarify that the activity surface is based on interpolation. The legend of figure 1 refers to a Tanimoto similarity of 0.69. While the authors use ECFP4 fingerprints later on, at this point it is not clear whether it is a Tanimoto similarity of ECFP4. The fingerprint should be specified.

Figure 4: Does not contain results for Tr^* , but results are reported in Table 1. For consistency, Tr^* should also be reported in Fig4 or a rationale given, why this was not done.

P15L685: 'k=0.6' : I assume this is should be $\mu_k = 0.6$ as described on P14L618

P19L838: 'We used 5 nearest neighbors and the mean similarity of the target dataset as the cutoff TC for each competing method for all subsequent network graphs'

It is not quite clear to me how the KNN graph was constructed. Does mean similarity refer to the mean similarity of all 5NNs? and where at most 5 edges drawn to the neighbors if their similarity exceeded the threshold? This should be clarified.

Figure 8: Flipping the depiction of structures C1 and C2 horizontally would align them better to C3-C5 making it easier to visually compare the structures.

Reviewer #4 (Remarks to the Author):

The authors have extended the results section of the manuscript with their revision. For instance, they have included more analyses about the TR's interpretability, which was one of the main highlighted benefits. With Fig.1 and the KNN graphs, applications of the method are clearer. However, there are still some concerns from my side, mainly related to lack of robustness of some explanations and statistical analyses, which were not corrected during the revisions.

- Overall, I find the Introduction a bit lengthy and difficult to follow. The presentation of the methods and state-of-art has become more unorganized, and I think there are small errors related to cheminformatics concepts or nomenclatures that have become more pronounced with the inclusion of additional techniques during the revision.

Pg. 1 line 70: "ECFP4 (...). The graph representations, on the other hand, envisions 2D chemical structures as graphs." This statement is incorrect because 'on the other hand' indicates that ECFP4 does not envision chemical structures as graphs, and it does.

Along these lines, it is not clear to me why "descriptor/FP" are in the same category and I think this also leads to confusing explanations below (see next 2 comments).

- Pg. 3 line 101: "When dealing with more interpretable descriptors, knowing the set of "most important" features can be used to provide chemists with relevant substructures for the desired activity, but they are typically inadequate for directly mapping to chemically valid and feasible molecules." Not sure what the authors mean with this. Feature importance scores from RF/SVM models can be mapped onto the 2D structure of the compound if FPs are used (e.g. Visualization and Interpretation of Support Vector Machine Activity Predictions | Journal of Chemical Information and Modeling). Authors should also mention this, since it seems that shallow learners do not enable mapping into a valid molecule, which is incorrect.

- Pg. 6 line 275: "We calculate folded ECFP4 descriptors using with Morgan fingerprints" This sentence does not make sense. ECFP and Morgan are fingerprints, normally used as synonyms. ECFP4 descriptors cannot be calculated with Morgan fingerprints..

- Pg. 2. Line 93: "Model interpretability (...) typically consists of computing feature importance scores, training data importance scores, developing locally interpretable models to approximate global black-box algorithms, and generating counterfactuals." This sentence is not clear to me. Not sure if the authors are enumerating different strategies? If so, what are "training data importance scores"? Could the authors add references for each of the mentioned methods?

- Pg.3 Line 103 - 'provide chemists with relevant substructures for the desired activity' The authors should add 'predicted' activity since the analyses of 'most important' features are done on the basis of a prediction model.

- Some acronyms are not defined the first time they appear in the text, e.g. SEA, CNN, ECFP

- Fig. 1 is presented and mentioned in the Introduction, but should be in Results

- Authors should cite previous work on metric learning, molecular kernels, or kernel regression in the QSAR field, as mentioned in R1.

- There are many statements indicating that some methods 'outperform' others, e.g. ensemble TR outperforms TCNN on scaffold split (Pg. 16 line 696). However, authors might be overinterpreting the results. 1) there are no statistical tests to confirm these statements, 2) even if some comparisons were significant, the differences in performance are generally quite small to conclude that. Figure 4 also shows how minor performance differences observed in Table 1 are not meaningful due to the variability across folds.

- Authors could add TR* to Figure 4 and move Table 1 to Supplementary or remove it, since is redundant.

Even though the presented approach and chemical space visualizations are interesting, and some results are compelling, I still find this work too preliminary for publication. I recommend authors to simplify the presentation, be more concise and reduce the broad statements. If the method is not improving state-of-art performance, authors could focus on highlighting interpretation benefits compared to the other methods.

Detailed Response to Reviewer Comments: Topological Regression in Quantitative Structure-Activity Relationship Modeling

Reviewer Comments

Reviewer #1

I have read the author's response to the reviewer's comments and it seems the most important topics are addressed. I don't have any urgent additional comments.

The authors would like to thank Reviewer No. 1 for appreciating the changes made in the revised manuscript.

Reviewer #2

The authors have effectively addressed all the comments from the reviewers and carefully revised the manuscript, alongside with providing now the code accessible in the platform GitHub. Therefore, the revised manuscript can be published in its current form.

Reviewer #2 (Remarks on code availability):

The code provides an appropriate README file with a detailed introduction, and the extent the results of the paper are reproducible is guaranteed.

The code, along with its scripts, is a usable resource for the QSAR community.

The authors would like to thank Reviewer No. 2 for appreciating the changes made in the revised manuscript.

Reviewer #3

*The authors would like to thank Reviewer No. 3 for their constructive review of the manuscript and valuable comments. Regarding the productive concerns raised by **Reviewer 3**, the following changes have been incorporated in the revised manuscript:*

The authors present a much improved version of their manuscript addressing all points raised by the reviewers. I consider the revised publication acceptable for publication.

A few minor issues should be fixed:

Figure 1: Shows a smooth interpolated activity surface based on the activities of the projected molecules. For clarity, the authors could clarify that the activity surface is based on interpolation. The legend of figure 1 refers to a Tanimoto similarity of 0.69. While the authors use ECFP4 fingerprints later on, at this point it is not clear whether it is a Tanimoto similarity of ECFP4. The fingerprint should be specified.

We thank the reviewer for bringing this up, the clarity has been improved by stating the activity surfaces were interpolated and that the calculated Tanimoto similarity was based on ECFP4 fingerprints.

Figure 4: Does not contain results for Tr^* , but results are reported in Table 1. For consistency, Tr^* should also be reported in Fig4 or a rationale given, why this was not done.

As suggested by Reviewer 3 and 4, we have included Tr^ in Figure 4 for completeness.*

P15L685: 'k=0.6' : I assume this is should be $\mu_k = 0.6$ as described on P14L618

We appreciate the reviewer bringing this to our attention, we have updated the manuscript to μ_k .

P19L838: 'We used 5 nearest neighbors and the mean similarity of the target dataset as the cutoff TC for each competing method for all subsequent network graphs'

It is not quite clear to me how the KNN graph was constructed. Does mean similarity refer to the mean similarity of all 5NNs? and where at most 5 edges drawn to the neighbors if their similarity exceeded the threshold? This should be clarified.

We have added more explanation to the paragraph to improve the clarity of how the KNN graphs were constructed. We have included that 'at most 5 connections would be established if their similarities were greater than the fixed cutoff TC.' We have also added that the mean similarity was calculated from the entire target dataset.

Figure 8: Flipping the depiction of structures C1 and C2 horizontally would align them better to C3-C5 making it easier to visually compare the structures.

We have flipped the structures of C1 and C2 horizontally so that all molecules are aligned which makes it easier to visualize the chemical's structural similarity.

Reviewer #4

*The authors would like to thank Reviewer No. 4 for their constructive review of the manuscript and valuable comments. Regarding the productive concerns raised by **Reviewer 4**, the following changes have been incorporated in the revised manuscript:*

The authors have extended the results section of the manuscript with their revision. For instance, they

have included more analyses about the TR's interpretability, which was one of the main highlighted benefits. With Fig.1 and the KNN graphs, applications of the method are clearer. However, there are still some concerns from my side, mainly related to lack of robustness of some explanations and statistical analyses, which were not corrected during the revisions.

- Overall, I find the Introduction a bit lengthy and difficult to follow. The presentation of the methods and state-of-art has become more unorganized, and I think there are small errors related to cheminformatics concepts or nomenclatures that have become more pronounced with the inclusion of additional techniques during the revision.

We thank the reviewer for the comment. We have tightened the introduction in the revised version and made modifications to improve the overall organization and clarity regarding the cheminformatics nomenclatures.

Pg. 1 line 70: "ECFP4 (...). The graph representations, on the other hand, envisions 2D chemical structures as graphs." This statement is incorrect because 'on the other hand' indicates that ECFP4 does not envision chemical structures as graphs, and it does.

Along these lines, it is not clear to me why "descriptor/FP" are in the same category and I think this also leads to confusing explanations below (see next 2 comments).

We thank the reviewer for pointing this out, we realize envisions is the wrong word choice, so we have replaced it with characterizes. The authors intent of this paragraph is to introduce readers to some of the common machine comprehensible molecular representations used in QSAR studies. While ECFPs are derived from the graph structures, they are characterized as vectors for machine comprehension, so envisions is not a proper word for what we are trying to convey. The main point of this sentence is to make a distinction between the mathematical graph structure vs vector-based descriptors and fingerprints. On this note, that is why fingerprints and descriptors were grouped together. To improve clarity for the readers, we have reworded the introduction of representations to be based on their machine comprehensible format, "Three commonly used representations are: (a) vectors such as classical molecular descriptors or molecular fingerprints (FPs), (b) graphs, and (c) strings such as Simplified Molecular Input Line Entry System (SMILES)."

Additionally, based on the commonly used definition of molecular descriptors from Todeschini and Consonni [1], "The molecular descriptor is the final result of a logic and mathematical procedure which transforms chemical information encoded within a symbolic representation of a molecule into a useful number or the result of some standardized experiment," fingerprints are often considered descriptors. For example, many of the cited work in the manuscript refer to fingerprints as a type of descriptor [3-5]. Thus, to improve clarity, we have followed Grisoni's, Ballabio's, Todeschini's and Consonni's distinction in [2] which separates them by the terms classical descriptors vs fingerprints.

[1] Todeschini, Roberto, and Viviana Consonni. *Handbook of molecular descriptors*. John Wiley & Sons, 2000.

[2] Grisoni, F., Ballabio, D., Todeschini, R., & Consonni, V. "Molecular descriptors for structure–activity applications: a hands-on approach." *Computational Toxicology: Methods and Protocols* (2018): 3-53.

[3] Cherkasov, Artem, et al. "QSAR modeling: where have you been? Where are you going to?." *Journal of medicinal chemistry* 57.12 (2014): 4977-5010.

[4] Kwon, Sunyoung, et al. "Comprehensive ensemble in QSAR prediction for drug discovery." *BMC bioinformatics* 20.1 (2019): 1-12.

[5] Balfer, J., Bajorath, J.: *Visualization and interpretation of support vector machine activity predictions. Journal of Chemical Information and Modeling* 55(6), 1136–1147 (2015)

- Pg. 3 line 101: "When dealing with more interpretable descriptors, knowing the set of "most important" features can be used to provide chemists with relevant substructures for the desired activity, but they are typically inadequate for directly mapping to chemically valid and feasible molecules." Not sure what the authors mean with this. Feature importance scores from RF/SVM models can be mapped onto the 2D structure of the compound if FPs are used (e.g. Visualization and Interpretation of Support Vector Machine Activity Predictions | Journal of Chemical Information and Modeling). Authors should also mention this, since it seems that shallow learners do not enable mapping into a valid molecule, which is incorrect.

We have removed the sentence and have included that interpretable fingerprints can be mapped back onto the molecules to visualize the substructure prediction contributions and cited multiple papers that highlight the ability of shallow learners. Related to the previous point, we have also mentioned that molecular interpretability is largely based on the interpretability of the underlying representation, and tried to make more of a clear distinction between classical descriptors and fingerprints as they can have differing interpretability. Previously, our intention was to argue that even if models produce feature importance scores that can help us acquire substructure contributions, isolating the substructures with greatest contributions may not always lead to feasible novel molecules or design ideas. We have clarified this argument in the revised manuscript. We have stated that 'although feature importance measures increase the explanatory power of machine learning models, caution must be taken when these scores are invoked on molecules outside the applicability domain of the model, as prediction importance does not always translate to biological relevance.'

This is demonstrated in [1] as they highlight the ability of prediction importance's to diagnose SVM's with different kernels. Additionally, in [2] they assign manual and automatic quality scores to the mappings, proxying trust in the mappings, which shows that different methods and molecular representations can fail to capture the biological relevant substructures for every molecule (e.g. Figure 7 in [2] where they show 'SMARTS patterns may fail to account for all of the biologically relevant knowledge'). Additionally, in [3] they mention that to avoid misinterpretation, the model must be predictive based on statistical validation and that interpretation should only be considered within the applicability domain of the model. However, as mentioned in [4] they find that some high predictive performance models failed to highly rank the important patterns and proposed an alternative measure of interpretation accuracy and benchmark datasets. Lastly, [5] analyzed the robustness of various atom coloring methods with different molecular representations and QSAR modeling methods and found that 'Even in ideal cases where the contribution of individual atoms is known, we cannot always recover the important atoms for some descriptor/method combinations,' and that the feature importance's are very sensitive to selection of descriptors and QSAR modeling methods.

[1] Balfer, J., Bajorath, J.: *Visualization and interpretation of support vector machine activity predictions. Journal of Chemical Information and Modeling* 55(6), 1136–1147 (2015)

[2] Marchese Robinson, R.L., Palczewska, A., Palczewski, J., Kidley, N.: Comparison of the predictive performance and interpretability of random forest and linear models on benchmark data sets. *Journal of chemical information and modeling* 57(8), 1773–1792 (2017)

[3] Polishchuk, P.: Interpretation of quantitative structure–activity relationship models: past, present, and future. *Journal of Chemical Information and Modeling* 57(11), 2618–2639 (2017)

[4] Matveieva, M., Polishchuk, P.: Benchmarks for interpretation of qsar models. *Journal of cheminformatics* 13(1), 41 (2021)

[5] Sheridan, R.P.: Interpretation of qsar models by coloring atoms according to changes in predicted activity: how robust is it? *Journal of chemical information and modeling* 59(4), 1324–1337 (2019)

- Pg. 6 line 275: “We calculate folded ECFP4 descriptors using with Morgan fingerprints” This sentence does not make sense. ECFP and Morgan are fingerprints, normally used as synonyms. ECFP4 descriptors cannot be calculated with Morgan fingerprints.

We thank the reviewer for mentioning this point and realize that the sentence needed improved clarity. We use RDKit for fingerprint extraction and RDKit only contains an implementation of the Morgan algorithm that allows the user to specify the radius, or stopping iteration, and the desired fixed bit size, which allows the computation of hashed ECFPs. To maximize the reproducibility and to improve clarity, we have altered the sentence to say: “We calculate folded ECFP4 fingerprints using RDKit’s implementation of the Morgan algorithm with a radius of 2 atoms and bit-size of 1024.” (we understand that Morgan and ECFP may be used synonymously, but ECFPs are a variant of the Morgan algorithm that allow quicker computation of fingerprints given a desired stopping iteration, or diameter/radius, whereas the Morgan algorithm continues iterations striving for canonicalization) [1]

[1] Rogers, David, and Mathew Hahn. "Extended-connectivity fingerprints." *Journal of chemical information and modeling* 50.5 (2010): 742-754.

- Pg. 2. Line 93: “Model interpretability (...) typically consists of computing feature importance scores, training data importance scores, developing locally interpretable models to approximate global black-box algorithms, and generating counterfactuals.” This sentence is not clear to me. Not sure if the authors are enumerating different strategies? If so, what are “training data importance scores”? Could the authors add references for each of the mentioned methods?

Yes, we are enumerating different strategies that are commonly used to interpret different modeling methods. We realize ‘training data importance scores’ is not a common term and can lead to confusion, so we have replaced it with ‘influence functions’ which trace model predictions back to its training data to identify training points most responsible for the prediction. We have included some common references for each strategy as well.

- Pg.3 Line 103 – ‘provide chemists with relevant substructures for the desired activity’ The authors should add ‘predicted’ activity since the analyses of ‘most important’ features are done on the basis of a prediction model.

We thank the reviewer for mentioning this, as suggested earlier, we have removed this sentence and have only included that the feature importance’s are based on the predictions.

- Some acronyms are not defined the first time they appear in the text, e.g. SEA, CNN, ECFP

We thank the reviewer for bringing this to our attention, we have defined the acronyms by their full names in the text where they first appeared.

- Fig. 1 is presented and mentioned in the Introduction, but should be in Results

We use figure 1 as motivation of metric learning and to show how it achieves smoother activity landscapes compared to other popular projection techniques. Since the success of metric learning's ability to smooth activity cliffs is an important part of the motivation and rationale behind the development of topological regression, we elected to keep figure 1 in the introduction.

- Authors should cite previous work on metric learning, molecular kernels, or kernel regression in the QSAR field, as mentioned in R1.

We thank the reviewer for this suggestion. We have incorporated citations for each of the mentioned methods. Specifically, we have included [1] which shows how metric learning smooths activity cliffs, [2-3] which are molecular kernels that are descriptor/fingerprint free methods of computing molecular similarities, and [4-6] as different kernel methods applied in QSAR modeling.

[1] Kireeva, N.V., Ovchinnikova, S.I., Kuznetsov, S.L., Kazennov, A.M., Tsvadze, A.Y.: Impact of distance-based metric learning on classification and visualization model performance and structure–activity landscapes. *Journal of computer-aided molecular design* 28, 61–73 (2014)

[2] Fröhlich, H., Wegner, J.K., Sieker, F., Zell, A.: Kernel functions for attributed molecular graphs—a new similarity-based approach to adme prediction in classification and regression. *QSAR & Combinatorial Science* 25(4), 317–326 (2006)

[3] Mohr, J.A., Jain, B.J., Obermayer, K.: Molecule kernels: a descriptor- and alignment-free quantitative structure–activity relationship approach. *Journal of Chemical Information and Modeling* 48(9), 1868–1881 (2008)

[4] Yamanishi, Y., Araki, M., Gutteridge, A., Honda, W., Kanehisa, M.: Prediction of drug–target interaction networks from the integration of chemical and genomic spaces. *Bioinformatics* 24(13), 232–240 (2008)

[5] Gajewicz-Skretna, A., Furuhashi, A., Yamamoto, H., Suzuki, N.: Generating accurate *in silico* predictions of acute aquatic toxicity for a range of organic chemicals: Towards similarity-based machine learning methods. *Chemosphere* 280, 130681 (2021)

[6] Jacob, L., Vert, J.-P.: Protein-ligand interaction prediction: an improved chemogenomics approach. *Bioinformatics* 24(19), 2149–2156 (2008)

- There are many statements indicating that some methods 'outperform' others, e.g. ensemble TR outperforms TCNN on scaffold split (Pg. 16 line 696). However, authors might be overinterpreting the results. 1) there are no statistical tests to confirm these statements, 2) even if some comparisons were significant, the differences in performance are generally quite small to conclude that. Figure 4 also shows how minor performance differences observed in Table 1 are not meaningful due to the variability across folds.

We agree with the reviewer and have modified the relevant statements. We submit that the Law of Parsimony works in favor of our posited TR method. Although the difference in predictive performance between TR and other competing models are small (as the reviewer points out), but our method is way less complex as compared to other methods because at its core TR is a two-stage procedure involving multivariate log-linear regression followed by k-nearest neighbor (the complexity of the parametric model is simply the number of estimated regression parameters and that for K-NN is the size of the set containing the anchor points in relation to the size of the training set). Consequently, if two models offer similar predictive performance, Law of Parsimony dictates that simpler model should be preferred.

We have clarified this argument in the revised manuscript.

- Authors could add TR* to Figure 4 and move Table 1 to Supplementary or remove it, since is redundant.

As suggested by the reviewer, we have added Tr to Figure 4 for completeness. However, using figure 4 alone would not allow the reader to see the exact numerical means of the different methods performance. We have elected to keep table 1 to provide readers with the exact average performance of each method.*

Even though the presented approach and chemical space visualizations are interesting, and some results are compelling, I still find this work too preliminary for publication. I recommend authors to simplify the presentation, be more concise and reduce the broad statements. If the method is not improving state-of-art performance, authors could focus on highlighting interpretation benefits compared to the other methods.

REVIEWERS' COMMENTS

Reviewer #3 (Remarks to the Author):

These authors have adequately addressed the issues raised by the reviewers and I support the publication of the manuscript in its current form.

Reviewer #4 (Remarks to the Author):

Thanks to the authors for carefully answering the previous comments and making changes to the manuscript. I would recommend the publication of this research paper.